# General practitioners and sales representatives: Why are we so ambivalent?

**Adriaan Barbaroux** [1,2]*, **Isabelle Pourrat**[1], **Tiphanie Bouchez**[1]

**1** Département d'Enseignement et de Recherche en Médecine Générale, RETINES, HEALTHY, Université Côte d'Azur, Nice, France, **2** LAPCOS, Université Côte d'Azur, Nice, France

* adriaan.barbaroux@univ-cotedazur.fr

**Data Availability Statement:** No - some restrictions apply There are ethical restrictions on sharing relevant data as data contain potentially identifying and sensitive information. For data access requests, researchers may contact the

## Abstract

### Introduction

Accepting gifts from pharmaceutical sales representatives (sales reps) or meeting them is correlated with excessive, more expensive and sometimes less rational prescribing. French general practitioners (GPs) tend to hold an unfavorable opinion of the pharmaceutical industry, yet the behavior they adopt with sales reps is generally favorable. Until now, no study has sought to explain the reasons for this discrepancy. This study explores GP experiences to better understand their ambivalent behavior.

### Method

This qualitative descriptive study was based on semi-structured face-to-face interviews with French GPs in the south-east of France. An interpretative phenomenological approach was chosen to explore individual professional practices and to model the phenomenon through in-depth analysis of semi-structured interviews. A general inductive analysis was carried out. Data were analyzed by researchers from different disciplines (psychology, sociology and general practice).

### Results

Ten GPs were interviewed for an average of 50 minutes. The analysis revealed three forces that combine to motivate GPs to keep meeting sales reps despite their unfavorable opinion of these visits: practical reasons such as the need for a substitute for continuing education; social and cultural reasons such as courtesy towards representatives; and psychological mechanisms such as cognitive dissonance and a hidden curriculum.

### Discussion

The GP-representative relationship is complex and involves psychological mechanisms that the medical profession often fails to recognize. GPs use reps as a convenient tool for continuing education, particularly in the setting of a private practice where GPs feel pressed for time. Cognitive dissonance is a well-supported theory in social psychology that explains how a person maintains a behavior while having an unfavorable opinion of it. Since GP meetings with sales reps start during their internship, they could also be considered as part

departement d'enseignement et de recherche en médecine générale, 28 Avenue De Vallombrose, 06107 Nice, France via laura.bruley@univ-cotedazur.fr. Researchers may also contact the corresponding author with data access requests.

**Funding:** The authors received no specific funding for this work.

**Competing interests:** The authors have declared that no competing interests exist.

**Abbreviations:** GP, General Practitioner; USA, United States of America.

of a hidden curriculum. The strength of this work is to combine medical, social psychological and sociological perspectives with the original interpretative phenomenological approach. When the veil is lifted on individual ambivalence, the questions raised are more social and political than individual.

## Introduction

In the United States of America (USA), the biopharmaceutical sector generates nearly $1.2 trillion in economic output annually when direct (biopharmaceutical industry revenue), indirect (vendor and supplier activity), and induced effects (additional private economic activity) are considered [1]. Financial relationships between the leaders of influential professional USA medical associations and industry are extensive: 72% of such associations have financial ties to industry [2]. The marketing of opioid products to physicians in the USA was associated with increased opioid prescribing and higher mortality from overdoses [3]. The pharmaceutical industry in France has one of the largest promotional budgets of any country [4]. The pharmaceutical industry represents 100,000 jobs, and with its 54-billion-euro annual turnover, is the fourth biggest contributor to the French Trade Balance (0.15% of the Gross Domestic Product). The role of pharmaceutical sales representatives (sales reps) is regulated by the government through a Sales Visit Charter and a certification procedure but, despite more stringent regulatory requirements than in North America, the information concerning drugs provided by sales reps to GPs remains poor [5, 6]. The Sales Visit Charter promotes good practices such as the presentation of approved product information including side effects and contraindications. Despite strict regulations (health authorities can impose fines of up to €10,000 or 10% of a product's annual sales revenue) [7], key provisions within the Sales Visit Charter, such as mentioning adverse effects, precautions and contraindications are rarely respected [5, 6]. Pharma sales reps are not allowed to provide content for continuing education courses. A national fund for continuing medical education finances free courses for GPs and compensation for the loss of three days' worth of medical income while training. This non-biased continuing medical education is mandatory [8]. Initial and continuing medical education are supposed to be provided by impartial experts but these experts can be opinion leaders with financial ties to the industry [9–12]. Despite these measures, the pharmaceutical industry still plays an active role in research and in general practitioner (GP) training from the preclinical years to continuing medical education [11]. The French National Health Authority, "Haute Autorité de Santé" (HAS), encourages professionals and health care facilities to address the issue of sales visits and provides tools such as a French translation of the World Health Organization's guide "Understanding and responding to pharmaceutical promotion: a practical guide". HAS also provides guidance to health care facilities on how to regulate sales visits [13, 14].

French primary care is mainly organized as a private, independent service, although financed by the public health system. Like most primary care providers in France, GPs mainly work in private practice and are paid on a fee-for-service basis. Group practice is now dominant among GPs and 81% of GPs younger than 50 declare that they work in a group [15]. Since the 2000s, teamwork in multidisciplinary primary care centers has been developed, mainly to maintain primary care services in low-income areas. In France, GPs meet an average of 5.6 sales reps per week, and spend an average of 55 minutes per week with them [16]. Nine out of ten GPs in France were offered a gift from a pharmaceutical company between 2011 and 2016 [17].

Accepting gifts from sales reps or meeting them is correlated with excessive, more expensive and less rational prescribing [18]. A recent study confirmed this finding about French GPs using indicators such as the prescription of antibiotics, benzodiazepines and vasodilators [17]. Free drug samples and gifts of food and drink in the office are not allowed in France, but GPs still receive gifts and invitations to restaurants since the law allows for exceptions. Promotional budgets could therefore be viewed as an investment that ensures an increase in the number of prescriptions [4, 17, 19]. A report by the French Inspectorate General of Social Affairs (*Inspection Générale des Affaires Sociales–IGAS*) studied the information GPs receive about drugs. The report concluded that, after the drug database *Vidal*, sales rep visits were the second most used source of information for practitioners, despite a negative opinion held against detailing visits [4]. This situation is problematic on a financial level (costs are indirectly born by the community via medication prices) as much as on a social and ethical level (citizens' demand for more transparency and more independent training for prescribers) and on a public health level because of potentially inadequate prescribing practices [4, 20, 21]. Therefore the independence of prescribers remains an essential goal [22].

In 2003, British GPs reported that they received visits from sales representatives for six major reasons: the convenience and availability of information about drugs; the legitimacy of reps as a source of information; out of courtesy; to receive gifts; social and intellectual interaction; to conform with organizational and cultural norms [23]. The reasons GPs gave for seeing reps were reported in "neutral" and "legitimate" terms, as GPs largely believed in their own immunity to commercial marketing strategies. Another qualitative study pointed out that French GPs' interactions with sales reps provided them with information that was easy to access and easy to use [24]. GPs also described a lack of confidence in their own pharmaceutical knowledge when leaving university [24].

These observations highlight the seemingly ambivalent position of GPs, who believe in their own ability to resist the influence of reps but not necessarily in the ability of other GPs [9]. They are well aware of the reps' commercial role and hold a rather negative opinion of them, describing reps as advertisers who are trying to influence them to prescribe in a less efficient way [3]. However, their behavior remains rather favorable towards the pharmaceutical industry, as they still predominantly receive sales rep visits (80% of practitioners in France) [11, 23, 25]. So far, no study has attempted to understand the ambivalence of GPs: why do so many of them receive sales rep visits even though they hold a rather negative opinion of the pharmaceutical industry? This study explores GP experiences to better understand their ambivalence.

## Method

### General design

An exploratory qualitative study was conducted based on a constructivist paradigm [26]. In this paradigm, qualitative research methods are considered as a means to allow the investigator to be the primary interpretive instrument and reality is considered as socially constructed and knowledge as a product constructed by people taking active part in a research process [26]. An interpretative phenomenological approach was used, that is, individual professional practices were explored and the phenomenon was modeled through the in-depth analysis of singular experiences, the meaning that participants gave to their experience of meeting sales reps, and the underlying psychological mechanisms [27]. The purpose of a phenomenological approach is mainly to shed light on the essence of a person's experience in relation to a specific phenomenon, in this instance meeting sales reps despite holding a negative opinion of them. In a phenomenological study, sample sizes are typically small because data saturation is not

essential if the objective is to understand the meaning individuals give to their own lived experience [28]. Semi-structured interviews with general practitioners were conducted face to face as appropriate for the intimacy of the attitude being explored. Each interview was recorded and transcribed verbatim. Field notes were taken during the interviews and immediately after.

Transcripts were analyzed one interview at a time to explore individual perspectives. Texts were commented on, then themes were coded, organized and finally formulated to identify common patterns shared by the different experiences [27, 29–31]. Results were interpreted by a multi-field team composed of three GPs (AB, IP, TB), a sociologist (IF), and a social psychologist (IM). The heterogeneity of the team's skills aimed at identifying a broader range of themes to enlarge the scope of the analysis.

The interviews were conducted at the GP's office by a male family medicine resident (AB) previously trained by members of the research team (IP, IF). The practitioners were asked to react to a series of three clinical scenarios through open questions (the interview guide is available in **S1 Appendix**). The clinical scenarios represented current situations encountered by general practitioners in their work and the information sources they can turn to. The interview guide was developed from existing literature [24] (identifying professional situations of uncertainty and reasons for meeting sales reps) and from two preliminary interviews. Both preliminary interviews were based on an interview guide with open questions relating to the relationship with sales reps ("What do you expect from a sales rep?" "Can you describe the way your last encounter with a rep went?"). Both practitioners interviewed showed signs of defense mechanisms related to social desirability [32]. Thus, the guide structure was modified to include the three clinical scenarios. The data from the preliminary interviews was retained in this study, in accordance with the standards in qualitative research [27, 29]. The modified interview was designed to elicit the same reflections and allow the GPs to address sales rep visits by means of the scenarios without requiring the investigator to mention them. Each participant discussed the three scenarios. The scenarios presented a short story describing GPs in a situation of doubt and the way they would react to it (for instance by seeking the advice of a fellow GP, or by prescribing a new drug discovered during a conference). This context was chosen because of previous evidence that in a situation of doubt a practitioner is the most sensitive to the influence of sales reps [9]. The scenarios served to gradually bring up the issue of sales rep visits without causing the reluctance and social desirability bias observed in the two preliminary interviews. The questions that followed the three scenarios were similar ("*What sources of information do you regularly use*?"–"*Can you tell me more about sales reps*?"). The next questions aimed at progressively focusing on the GP-rep relation ("*Have you ever considered no longer meeting reps*?"–"*I am particularly interested in the apparent contradiction between what you tell me*" . . ."*and, at the same time,*" . . ."*How did you come to this opinion*?"). Some questions were not directly related to sales reps in order to avoid the defensive reactions observed in the preliminary interviews. The second and the third scenarios only served to renew the discussion when necessary by moving on to the next scenario. The first scenario concerned the prescription of sitagliptin to a frail patient, the second the switch to oral anticoagulation in an elderly and poly-medicated patient and the third the prescription of rivaroxaban to a young patient with paraphlebitis, following the advice of a colleague.

## Population and recruitment

A convenience sampling method was used to select people with a range of experience in meeting sales reps. No specific inclusion or non-inclusion criteria were imposed since the research team considered that any GP could contribute new data to explain the phenomenon studied. The selection of practitioners was based on variability criteria for the following characteristics:

age, gender, rural location, years of practice, patient list size, single or collective practice, and paid subscription to a journal or membership in a peer group (see chart no. 1). Peer groups are structured small groups of 6 to 12 health care professionals who meet regularly to reflect on and improve their practice. They also serve to develop professional networks. They are very popular in France, where participation can give rise to a government payment to members when the group is accredited (registered and recognized by the government) [33]. These characteristics were expected to lead to a diverse sample of experiences and were based on the literature describing how the GP-rep relationship depends on social norms, age, patient list size and the type of continuing medical education [4, 24]. Participants were asked about these characteristics at the end of the interview.

The GPs' phone numbers were found in the phone book. They were recruited in a coastal city with a population of more than 300,000 people in the south of France and in three neighboring towns of 1,300–7,000 people. The GPs were asked to answer a survey as part of an end of course medical thesis. The survey was presented to them as dealing with the information sources used by general practitioners, without mentioning promotional detailing visits. Ten GPs declined to participate because of a lack of time.

## Validity and reliability

The research protocol and the presentation of results are based on COREQ criteria (COnsolitaded criteria for REporting Qualitative research), see **S1 Checklist** for further information. For confidentiality reasons, some of the information such as the gender of interviewed participants is not presented and no personal characteristics such as gender that might identify a speaker are attached to specific quotes.

Written consent was obtained from each participant. Ethics approval was not required because of the non-pharmaceutical bio-medical nature of this research [34]. Ethical aspects nevertheless respect the French legislation for data collected before January 2016. The authors did not receive any funding for this study.

This research article is part of the thesis work of a male general medicine resident who was in charge of collecting data. This main investigator studied reference material [29–31, 35–38] beforehand to gain knowledge in qualitative research, and obtained the assistance of researchers in social psychology, sociology and general practice who formed a workgroup specifically for this study. The main investigator examined his own assumptions, and his reasons and interests for studying the research topic by brainstorming with the research team before the research started.

Thematic extraction was triangulated to discuss and enrich interpretation from different perspectives: all interviews were analyzed first by the principal investigator, then by another researcher (GP, psychologist, or sociologist). The analyses were convergent (no disagreements arose between the surveyors) and complementary (they helped identify new themes). The research team discussed the data sample and considered that it gathered a wide range of experiences about relationships with sales reps and no new theme emerged from the material analyzed. Theoretical validation was obtained by comparing the results with existing scientific data (see Discussion).

The relevant datasets are available on request to the corresponding author.

## Results

The material is composed of 10 interviews carried out from April to December 2014, of an average duration of 50 minutes (median of 46 minutes), for a total duration of 8 hours and 22 minutes.

The ten practitioners were 30 to 59 years old, and almost all of them charged the approved National Health rate. Two of them were women. Six of them worked in an urban environment; the other four worked in rural or semi-rural areas. One practitioner no longer received detailing visits. Two GPs received one sales rep visit a day, and one met with ten a week. Only one of them had a paid subscription to an independent journal, four of them declared reading free journals, and three took part in peer groups. These characteristics are summed up in Table 1.

All but one GP spontaneously mentioned sales reps when discussing the scenarios. When the physician did not address the issue spontaneously, the interviewer raised the subject specifically. The three scenarios did not each elicit a different pattern of responses, which was expected since the research question was not about the drugs discussed in the scenarios, but about the GP-rep relationship. Transcripts, or excerpts of responses, can be grouped into the following categories: motivations, physicians' perceptions, and values.

## Motivations

Here, "motivations" refer to the reasons GPs gave to explain why they meet sales reps.

The need for information was the motivation most often mentioned by practitioners to justify receiving sales rep visits despite their negative opinion about these interactions. The visits were described as *practical*, *approachable*, and sometimes of a *good quality*. This practicality justified the preference of sales rep visits over other information sources. "*What's important for the doctor is for it to be fast, concise, and straight to the point.*"

The participants attributed to rep visits a reassurance effect regarding their knowledge, and a "*starter*" effect as the first link of the information chain: meeting a sales rep was said to trigger more extensive research.

A social role was also attributed to sales reps, by providing a human relationship with the representative (felt like a "*break*") and by creating a medical "*network*" through sponsored meetings.

Professional constraints were also mentioned: "*It's true that I can't attend FMC* [Formation Médicale Continue–*Continuing Medical Education*] *as often as I'd like, so I'll admit that I enjoy having reps come [to my door].*"

## Physicians' perceptions

Here, "physicians' perceptions" refers to the perceptions and mental pictures that practitioners have of detailing visits and of their alternatives.

**Table 1. Demographics and practice characteristics of the ten GPs interviewed.**

|  | (Mean ± SD [Range]) |
| --- | --- |
| Age (years) | 48,8 ± 9 [30–59] |
| Years in practice | 15 ± 11 [1–34] |
| Enrolled patients | 906 ± 439 [400–1500] |
| Hours p. week workload | 55 ± 8 [43–65] |
| Number of sales reps per week | 3 ± 3 [0–10] |
|  | (n—percentage) |
| Urban setting | 6 (60%) |
| Group practice | 5 (50%) |
| Reads commercially-produced free GP journals | 4 (40%) |
| Subscribes to a paid medical journal | 1 (10%) |
| Member of GP peer group | 3 (30%) |
| Patients charged extra fees | 1 (10%) |

The physicians' perceptions of the sales reps were lukewarm, going from "*It's a lot of non-sense, they're selling that as if they were selling socks*" to "*he's well trained*", "*bulletproof*", "*well-oiled.*" Interviewed GPs raised few safety concerns about meeting reps: "*A few years ago, they had become the only source of information. I mean that those who didn't read 'La Revue Pre-scire', they'd prescribe; they followed the information brought by the visitors. It's dangerous to do that! But that's changed; it's changed a lot.*" None of them talked about the sales visit charter or drug-specific safety scandals.

A detailing visit was considered as an actual training opportunity. It could play the role of a "*ready-to-use training*", "*one symptom, one solution!*". Interviewed GPs described meetings with sales reps as "*a habit acquired at the hospital*". In comparison, all other training sources made available to practitioners were described as "*time-consuming*" and unsuited to their needs.

For some doctors, detailing visits were even considered essential: "*Otherwise, who would inform you?*" The diversity of available drugs seemed to create and maintain a feeling of dependence on the sales reps: "*As soon as he'd gone, I felt so useless!*"

The pharmaceutical industry's persuasive force was described as belonging to a past era because current sales rep budget restrictions mean fewer reps and gifts: "*There's no more! There's nothing left!*"

Participants described both the "commercial" aspects of the rep visits and the strategies that could be used to deal with them: combining sources of information, limiting the number of sales reps, and choosing reliable contacts. These strategies seemed to give them a feeling of control: "*When you increase your critical thinking, you're well prepared to attend any training session, you are equipped to deal with promotions.*"

## Values

By *values*, we mean what is considered to be true, right, good, and a worthy goal to be reached, something to stand up for. The values identified as conducive to maintaining an ambivalent behavior towards sales reps were independence, courtesy and pragmatism.

Independence seemed to be a value very strongly held by the interviewed practitioners. This could have motivated them to accept detailing visits in order to distance themselves from any kind of authority that requires them to maintain a degree of independence towards sales reps: "*I do what I want!*"

Courtesy is the set of rules and good manners that govern behavior in society. It was very often referred to as a reason for receiving sales rep visits: "*They've been waiting all day long,*" "*I always let them in; they've been waiting like everybody else!*"

Pragmatism seems to lead some GPs to grant more importance to sales reps and the experience of fellow GPs than to information drawn from studies or recommendations.

## Mechanisms involved in dealing with ambivalence

The data suggested significant sociological and psychological mechanisms that can help explain the reasons for the ambivalence of GPs towards the pharmaceutical industry: an experience-based rationale, self-efficacy and cognitive dissonance.

**Experience-based rationale.** To practitioners, empirical knowledge seemed more accessible and more real than scientific data. "*What's important to me is my results with my patients on a clinical and biological level [. . .] After that, they can show me their studies, their gizmos; I couldn't care less!*" Within this context, the GPs downplayed the poor quality of information provided by sales reps and considered their own practical experience to be sufficient to confirm the effectiveness of and tolerance for new drugs. Sales reps seemed to cultivate this

empirical view of medical expertise by using encouraging words such as "*Try it and you'll see!*". GPs' preference for tangible information is well known and seems legitimate, but these results show that an experience-based rationale can sometimes take precedence over scientific rigor. This may contribute to practitioners' acceptance of sales reps, as the relationship reinforces GPs' confidence in their own practical experience. Representatives can therefore become the first link in the chain of information, despite their commercial role, since information can subsequently be verified in the field.

**Self-efficacy.**   Several GPs declared being relatively inactive in the search for information about drugs, and claimed to feel unable to do so without detailing visits ("*If the reps won't talk about [drugs] with us, who will?*"; "*I can't attend [Continuing Medical Education] as often as I'd like, so I'll admit that I enjoy having reps come [to my door]*"). This phenomenon can be analyzed through the concept of self-efficacy as described by psychologist Albert Bandura in 1977 [39]. Bandura defines self-efficacy as the sense a person has of their level of ability to achieve a given task. Without this feeling of self-efficacy, individuals tend to adopt a passive behavior, or even quit, while individuals who have a high sense of self-efficacy adopt a pro-active behavior. This well-known concept aligns with our observations in this study and could help explain why some GPs rely on detailing visits. Meeting pharma reps seems to act as a substitute for continuing medical education because of the GPs' inability to use the complex and time-consuming scientific literature. This phenomenon seems to be compounded by the diversity of the pharmaceutical supply: "*As for me, I'm absolutely unable to know the whole Vidal [a French drug database] by heart!*" It could also be exacerbated by the fact that French GPs are often isolated and overworked [40].

**Cognitive dissonance.**   Changes made to the interview guide because of GP defense mechanisms observed in the initial interviews reveal that the GP-rep relationship is somehow taboo. The word *taboo* is used here in its common sense which is to express any ban related to a fact or the mention of it, without being limited to spirituality or religion. This taboo aspect has appeared in previous research: when the Ipsos French Health Institute surveyed GPs about representative visits, the investigators also had to "disguise" the true motivation of their study because too many GPs refused to answer. [16] By avoiding any reference to sales rep visits, GPs deny the discrepancy between their actual behavior towards sales rep visits and their opinion of them, thus avoiding unpleasant situations as well as self-questioning ("*I never said anything to anyone but I think that they're starting to change their practices [. . .] but I too was criticized [. . .] when I followed training programs here, I was the only one to ask questions, I was the only one to criticize*"). This "taboo" concept thus largely contributes to explain why so many GPs keep on receiving sales rep visits while expressing an unfavorable opinion about them. American psychologist Leon Festinger describes cognitive dissonance as the unpleasant tension that an individual feels when faced with two cognitions that are incompatible with each other (i.e., meeting sales reps and holding a negative opinion towards the pharmaceutical industry) [41]. In the presence of this tension, individuals use unconscious means to restore balance: by changing their behavior or changing their opinion. In a counter-intuitive manner, studies show that it is easier to modify one's opinion than to change one's behavior (for instance, after a failed attempt to stop smoking, smokers will tend to overstate the social benefits of smoking or to de-emphasize their will to stop) [41]. During the interviews, several GPs first expressed a very firm opinion towards sales reps: "*We feel a bit like prey, [receiving a pharmaceutical representative] is never a trivial matter*" or "*Don't count on the reps for information.*" Their speech became much less strong as the interview went on: "*Actually, when you increase your critical thinking, you're well prepared to follow any training session.*" An adjustment of the expressed opinion can be observed here that makes it more coherent with the behavior, which is to accept sales rep visits. This phenomenon can be observed within the framework of an

interview but also within the framework of a whole career: "*It used to be that I wasn't very keen on reps. I was very cautious with them [. . .] Anyway, it changed, it changed a lot.*" The taboo surrounding detailing visits is also associated with a feeling of cognitive dissonance: avoiding the topic helps to avoid the emergence of discordance between opinion and behavior.

## Discussion

The analysis of GPs' motivations, perceptions and values revealed mechanisms that contribute to the understanding of the ambivalence between opinions and behaviors towards sales reps: cognitive dissonance, experience-based rationale and the use of sales reps as a substitute for scientific literature, which can be linked to a lack of self-efficacy. The GP-rep relationship can be considered as a part of a continuum: the GPs interviewed described meeting with sales reps as something mandatory during initial training. Therefore, meetings with sales reps can be considered as social learning and part of the hidden curriculum. Lempp and Seale defined the hidden curriculum as a set of influences that function at the level of the organizational structure and culture and that include, for example, implicit rules individuals must follow to survive in the institution, such as customs and rituals, as well as aspects taken for granted [42]. Six learning processes belonging to the hidden curriculum of medical schools have been identified: loss of idealism, adoption of a "ritualized" professional identity, emotional neutralization, change of ethical integrity, acceptance of hierarchy, and the learning of less formal aspects of "good doctoring." Together they contribute to the enculturation of students as they become practitioners and members of the medical profession. Therefore, meetings with sales reps can be considered as a part of the hidden curriculum, which is maintained by the forces described above (practical reasons, substitute to training, cognitive dissonance, etc.).

The cognitive dissonance theory is well established and has been proven within the population at large but it has rarely been studied among GPs in France. This study brings new data suggesting that the mechanisms used to reduce cognitive dissonance may contribute to the GPs' ambivalence. An American qualitative study published in 2007 suggested that GPs felt cognitive dissonance when meeting sales reps [43]. The authors described psychological processes of denial and rationalization used to reduce this state of cognitive dissonance: avoiding thinking about the conflicts of interests, denying the fact that the relations with pharmaceutical companies have an impact on GP behavior, denying their responsibility, itemizing techniques used to remain unbiased and claiming that meeting sales reps brought a benefit to patients. While this study was carried out in a cultural and societal context very different from the French one, the existence of another study identifying signs of cognitive dissonance brings external coherence to our own results.

Another qualitative study in 2007 in the USA explored the reasons why prescribers from various health professions continued to interact with pharma sales reps [44]. Most prescribers believed that overall sales rep interactions were beneficial to patient care and health practice. They either trusted the information provided by sales reps or felt that they were equipped to evaluate it independently. Despite acknowledgment of study findings to the contrary, prescribers stated that they were able to effectively manage interactions with reps such that their own prescribing was not adversely impacted. These data were consistent with our findings and interpretations.

In 2003, a British study explored the reasons why GPs received sales rep visits. Even though the societal context is very different from ours, [23] the similarities are startling. The British study revealed six major reasons for receiving sales rep visits and all of them were spontaneously cited by the GPs we interviewed. Some of their observations are surprisingly similar to ours: "*Just because I have a pen with the name of a drug on it doesn't mean I'm going to prescribe it*" (England, 2003). "*It's not the pen that writes the prescription, it's me!*" (France, today). Or:

"*If it wasn't for the drug reps, we'd be left high and dry*" (England, 2003). "*If the reps won't talk about it with us, who will*?" (France, today). The British study also revealed that sales reps fill the need for continuing medical education: "*Vioxx, I did prescribe based on what I was being told by the rep. [. . .] I did perceive that I had a particular need for a new alternative.*" However, contrary to the results of the British study, French GPs did not use "neutral and legitimate" words to speak about detailing visits. Most of them seemed embarrassed to talk about it, suggesting a shift in perspective. The fact that "*times have changed*" was raised in every interview of French GPs confirms that sales reps are fewer than before, and receiving their visit is considered as less legitimate. Several French GPs said that sales reps were fewer and brought gifts less often. Other observational studies have arrived at the same conclusion [25, 45].

In contrast to a study published in 2018, [46] physicians talked relatively little about regulations and did not talk about drug-specific safety scandals such as those regarding rofecoxib (Vioxx) and benfluorex (Mediator). This difference is likely due to the interview guide, which sought to explore the reasons for the discordance between attitude and behavior and did not encourage these themes to emerge. Likewise, physicians did not talk about the Sales Visit Charter, suggesting that it remains relatively unknown more than 10 years after its adoption. Instead, they described a decrease in the frequency of medical visitors and in the size of the gifts offered to them. This decrease in the number and volume of gifts is confirmed by the analysis of the national "transparence-santé" database [45]. This decrease was used by doctors as an argument to trivialize the impact of sales rep visits on their practice. Although we can welcome the decrease in gifts, this trivialization may contribute to ongoing GP ambivalence towards sales visits.

## Strengths and limitations

Participants were not asked for any feedback on the transcripts or the analysis because of the sensitive aspect of the research topic. Due to the ambivalence of the doctors on this subject, time was required in the interviews to build trust and to gain in-depth understanding. It was therefore decided to give priority to the depth of the interviews rather than to the number of interviews. The study's design did not allow the authors to collect and incorporate data not known by the interviewed doctors, for instance the fact that in France GPs are quite isolated compared to Dutch ones [47]. This could have reinforced the feeling young doctors had of lacking knowledge and their need to rely on the information provided by the reps. Despite the method's limits, this work is consistent with the exploratory goal. The impact of the personal values of the main investigator on the data collection process was controlled by a group identification of his preconceived cognitions and by training the investigator in semi-directive interview techniques. Their impact on the interpretive process was limited by a double analysis of every interview and by the diversity of approaches provided by the cross-disciplinary team. The fact that all the interviewed doctors were from the southeast of France could be considered a limitation, but our results are consistent with the literature which brings external coherence. Moreover, the concepts that emerged are universal and therefore likely to be transposed to other settings.

This work's strength is that it is the very first exploratory study that seeks to understand the reasons why general practitioners are ambivalent towards sales reps in France. It is also of methodological and epistemological interest through the confluence of sociology, psychology and medicine. It could be useful for physicians, teachers and medical students to be aware of these results if they want to adopt a consciously chosen professional stance on pharma sales reps.

## Conclusion

This work suggests three types of possible explanations to understand why some GPs receive sales rep visits while holding a negative opinion of them. The first is practical, such as to obtain

information about new drugs, for convenience, and for social bonding. The second type of explanation is more social or cultural and is to remain within accepted codes of inter-personal relations and personal or professional characteristics (politeness, courtesy, welcoming the other person). When GPs lack confidence in their own experience and judgment capacities, when they experience personal or professional difficulties and are not active in searching for continuing education, sales reps are able to position themselves as information carriers. The third type of explanation concerns psychological factors such as cognitive dissonance, which is a well-established social psychology theory, and which helps understand how individuals whose behavior is not compatible with their opinions maintain a feeling of mental balance.

These observations lead us to consider the hypothesis that meetings with sales reps during initial training shape the GP-reps relationship which is then maintained over time by the three broad explanations described above. It has been demonstrated that the prohibition of gifts or samples has a positive influence on prescribing practices well over several years after students leave the university [48, 49]. These data bring external validity to our hypothesis. In 2017, Scheffer et al. ranked French medical universities according to their policy for managing conflicts of interest. At that time, the authors stated that French universities were behind a large number of countries, [50] despite some local initiatives and the willingness of French students to learn [21, 51].

Studies showed that this topic was still not adequately addressed by medical authorities, even though ten years have passed since the benfluorex scandal which led to a profound crisis in the French health sector [52–54]. Some changes in the initial training could be introduced to incorporate our findings and previous suggestions: prohibiting professors from receiving gifts from sales reps, nudging them towards greater independence, providing a basic curriculum for all medical students, developing and supporting existing awareness actions for students and health professionals in France or elsewhere [21, 55–58]. Continuing medical education could also be enhanced. GPs effectively felt the lack of free, non-biased and easy-to-access educational options that match their organizational constraints. French continuing medical education still suffers from a lack of independence and transparency in conferences and clinical practice guidelines despite the legal obligation for experts to disclose competing interests [44, 59]. The development of convenient, non-biased, and individual continuing medical education is therefore a public health necessity. Developing and valuing the independence of experts who speak at conferences and in learned societies is also necessary for the development of a culture of independence in health [2, 45, 60].

Previous studies raised the issue of inadequate application of the legislation in France, [7] which could explain why the GP-industry relationship is similar in France, Canada and the USA despite regulatory differences. In the French context, in which individual GPs are often overwhelmed by their working conditions and in which complex psychological factors are at work, GP-rep relationship issues may be more societal and political than individual.

## Supporting information

**S1 Checklist. COREQ checklist.**
(PDF)

**S1 Appendix. Interview guide.**
(DOC)

## Acknowledgments

First, the authors thank the study participants. They also thank the reviewers and the editor for valuable comments that helped to improve the manuscript, the Office of International

Scientific Visibility, Université Côte d'Azur, for proofreading, Catherine Regent (PhD) for proofreading and editing, and Pr. Jean-Baptiste Sautron, (GP) Pr. Isabelle Milhabet (IM, psychologist), Pr Isabelle Feroni (IF, sociologist), Dr. Carol-Anne Boudy (GP), Dr. Nicolas Hogu (GP) and Kamel Tribeche (clinical psychologist) for their proofreading and their critical advice.

## Author Contributions

**Conceptualization:** Adriaan Barbaroux, Isabelle Pourrat, Tiphanie Bouchez.

**Data curation:** Adriaan Barbaroux.

**Formal analysis:** Adriaan Barbaroux, Isabelle Pourrat.

**Investigation:** Adriaan Barbaroux.

**Methodology:** Adriaan Barbaroux, Isabelle Pourrat, Tiphanie Bouchez.

**Project administration:** Adriaan Barbaroux.

**Supervision:** Isabelle Pourrat, Tiphanie Bouchez.

**Validation:** Adriaan Barbaroux.

**Writing – original draft:** Adriaan Barbaroux, Isabelle Pourrat, Tiphanie Bouchez.

**Writing – review & editing:** Adriaan Barbaroux, Isabelle Pourrat, Tiphanie Bouchez.

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
