## [Decision Letter · Decision Letter 0]

23 Mar 2021

PONE-D-21-04814

GPs and pharma reps: why are we so ambivalent?

Crossing perspectives from sociology, psychology and general practice toward a qualitative study in France.

PLOS ONE

Dear Dr. Barbaroux,

Thank you for submitting your manuscript to PLOS ONE. After careful consideration, we feel that it has merit but does not fully meet PLOS ONE’s publication criteria as it currently stands. Therefore, we invite you to submit a revised version of the manuscript that addresses the points raised during the review process.

This article addresses an important issue, the relationship between primary care physicians and pharmaceutical sales representatives. The discussion is thoughtful and the focus on cognitive dissonance resonates with other related research. Currently, inadequate information is provided on methods (selection of participants, study conduct, and approach to analysis) to adequately assess the article's contribution to the existing literature on this issue. Additionally, the added French context needs to be considered, as is outlined below, given the extra restrictions on sales visits in France as compared with most countries. This was a key theme emerging in a previous qualitative study (Reynolds et al. 2018 - full ref below). As is noted below, given that three case scenarios with very different questioning afterwards were included in this study, reporting on how doctors responded to these scenarios and links to their views on sales visits need to be made more explicit. This is likely a valuable component of the study data and it is unclear why it was omitted from results reporting. Additionally, I had concerns about adequacy of assessment of data saturation with 3 different scenarios & questioning and 10 interviews in total. You may want to reconsider whether additional data are needed as well as a re-analysis. Here are my detailed recommendations:

Clarification of methods and ensuring that there is an explicit link between how methods are described and how the analysis is conducted and results reported and discussed. Dr Grundy describes this in detail in her comments, specifically in relation to the theoretical framing of the study and both the need for a more explicit description in the methods, with citations, and a clearer link between this framework and how results are analysed and presented. In addition the approach to selection of study participants (convenience, snowball, purposive?), how many were contacted and agreed or refused needs to be reported in greater detail? Were there specific inclusion criteria (for example that the interviewed GPs saw reps regularly)? If the aim was to explore effects of specific factors linked to demographics, for example gender differences, it’s unclear why only 2 female GPs were included, given a 44% female GP workforce in France, and also whether differences by demographics (age, gender) and number of years in practice or type of practice were considered as factors in assessment of whether saturation had been reached. It is also unclear when the interviews were carried out. This needs to be stated directly.

How the case scenarios informed the discussion and interpretation of results needs to be better reported. A study strength is the piloting of the approach used to questioning, and the introduction of revisions to avoid a social desirability bias and/or defensive responses to questioning. As these are two separate However, it is unclear how the three case scenarios were incorporated into this study, especially given the fundamental differences in approach to questioning following each scenario. The description of these scenarios in the article remains general and does not highlight these differences, nor include even a brief description of the drug and context per scenario. Were the specific drugs selected based on lack of evidence of therapeutic advantages or cost-effectiveness ratings? How did doctors respond to the specific scenarios? And were there any clear links between their responses and attitudes and experiences towards sales representatives? It is also unclear how these scenarios functioned in practice – did all of the doctors spontaneously mention sales representatives in response to these scenarios on a new drug prescription, or did they mention other factors more often and researchers introduced the link to sales representatives? Additionally, why were three separate scenarios used with different associated questions – some more directly related to sales representatives than others - and how often was each one used? When data saturation was being considered, how did the use of multiple scenarios affect judgments of thematic saturation?   

The study could be much better informed by the unique position of GPs in France in relation to reps as compared with most other countries, in that free medication samples and gifts of food and drink within the office are not allowed, and an attempt has been made to ensure information quality through public oversight of a certification procedure. Additionally, La Revue Prescrire used a network of GPs to analyse the content of sales visits in France over a 15-year period. (La Revue Prescrire 2006; 26(272):383-389) The context and French policies such as the sales visit charter would be expected to have some influence. How was the approach to questioning and analysis informed by this context? Also please include a description of the current approach to oversight. See for example: https://www.has-sante.fr/jcms/c_1099657/fr/visite-medicale-accompagner-les-professionnels-et-les-etablissements; Habibi et al. Journal of Law, Medicine and Ethics. 2016; 44: 602-613; Reynolds et al. Health Policy 2018; 122(3): 250-255; Mintzes et al. JGIM 2013; 28(10): 1368–1375. In the focus groups described in Reynolds et al., we also found that a strong influence from the Mediator scandal in how the French GPs discussed sales rep visits. This is likely to have been influenced by timing of the focus groups but an open question is whether in the current interview study there was any spontaneous discussion of safety concerns and/or of Mediator.

As noted by Dr Lexchin, editing of the manuscript by an English-language editor would be helpful. There are a few repeated “false friends”. For example ‘laboratoires’ translates to pharmaceutical companies, not to labs. In English a laboratory or lab is a facility where scientific experiments of all types take place. A ‘lab scientist’ is someone using usually very expensive specialised equipment (in their lab) to carry out scientific experiments.

We look forward to receiving your revised manuscript.

Kind regards,

Barbara Mintzes

Academic Editor

PLOS ONE

Journal Requirements:

2. Please consider changing the title so as to meet our title format requirement (https://journals.plos.org/plosone/s/submission-guidelines). In particular, the title should be "Specific, descriptive, concise, and comprehensible to readers outside the field" and in this case it could be preferable to avoid undefined acronyms

"NO - The funders had no role in study design, data collection and analysis, decision to

publish, or preparation of the manuscript."

Reviewers' comments:

Reviewer's Responses to Questions

**Comments to the Author**

1. Is the manuscript technically sound, and do the data support the conclusions?

Reviewer #1: Yes

Reviewer #2: Yes

Reviewer #3: No

2. Has the statistical analysis been performed appropriately and rigorously? 

Reviewer #1: Yes

Reviewer #2: N/A

Reviewer #3: N/A

3. Have the authors made all data underlying the findings in their manuscript fully available?

Reviewer #1: No

Reviewer #2: Yes

Reviewer #3: No

4. Is the manuscript presented in an intelligible fashion and written in standard English?

Reviewer #1: No

Reviewer #2: Yes

Reviewer #3: No

5. Review Comments to the Author

Reviewer #1: This is a qualitative study looking at why French GPs see sales representatives even though they generally have negative attitudes about the pharmaceutical industry. This type of inquiry is useful in formulating strategies about how to minimize the contact between these two groups and as such the authors should put forward policies towards that end based on their research.

1. The English in many places is awkward and the entire manuscript should be copy edited by someone who has English as a first language.

2. It would help in commenting on the manuscript if the lines were numbered.

3. In describing how interactions with sales representatives influence prescribing the authors should cite some of the literature coming out of the use of the Open Payments database.

4. Page 4: I think that it's highly likely that a portion of the medical profession is familiar with the terms cognitive dissonance and hidden curriculum and this sentence should be modified so as not to imply that it applies to the entire profession.

5. Page 4: The part of the sentence "complete the latter in a liberal field" is not clear.

6. Page 5: The authors should give a bit of background about general practice in France - how are doctors paid, do they generally practice alone or in groups, is there any funding for continuing medical education, do they also work in hospitals, etc.

7. Page 5: USA should be spelled out in full the first time it is used.

8. Page 5: The regulation of the activities of pharmaceutical representatives should be briefly described along with any penalties for violating the regulations.

9. Page 5: In what ways does the industry take part in the training of GPs?

10. Page 6: The authors should cite some of the literature showing that GPs believe in their own ability to resist being influenced by representatives but not necessarily in the ability of other GPs.

11. Page 6: There should be a reference to the literature describing the phenomenological approach.

12. Page 7: The existing literature used to generate the interview guide should be cited.

13. Page 7: What GP details are the authors referring to? The city where the study took place is mentioned later but should be mentioned here. How did the authors determine if the GPs saw sales representatives? Did all GPs invited agree and if not how many refused?

14. Page 9: I'm not sure how to interpret the final sentence in the Methods section.

15. Page 9: How does peer group continuing education function in France?

16. Page 10 (Table 1): What does "setting as a team" mean in the French context?

17. Page 10 (last paragraph): The first sentence is a repeat of what was said in the Introduction and Methods and should be deleted.

18. Page 11: What does "counter-phobic effect" mean?

19. Page 12: What does “training session” mean?

20. Page 15: The part of this paragraph starting "Thus, these data..." is a conclusion and belongs in that section.

21. Page 17: What “tendency” are the authors talking about?

22. Page 19: AMSA stopped publishing rankings in 2016.

Reviewer #2: Dear authors,

Thank you first for this interesting and useful work. I used the Côté-Turgeon assessment grid (https://www.pedagogie-medicale.org/articles/pmed/abs/2002/02/pmed20023p81/pmed20023p81.html )

to evaluate your article and all the points listed got a yes, there are so only minor corrections to do according to me.

1) I would rephrase this extract like following :

Many studies show a correlation between receiving a gift from pharma reps or meeting them

and an amount of bigger, more expensive and sometimes less rational prescriptions [give here the corresponding references, this one should belong to them I think: https://bmjopen.bmj.com/content/7/9/e016408 ]. A recent study published in the BMJ confirmed this for French GPs through indicators such as prescription of antibiotics, benzodiazepines and vasodilators [4]

2) This situation is as much of a problem on the financial level (with costs that are indirectly born by the community via medication prices) as it is on the social, ethical and Public Health levels. [2]

You could add this more recent reference here : Public health and the interests of the pharmaceutical industry: how to guarantee the primacy of public health interests? European Parliamentary Assembly, 2015 : http://semantic-pace.net/tools/pdf.aspx?doc=aHR0cDovL2Fzc2VtYmx5LmNvZS5pbnQvbncveG1sL1hSZWYvWDJILURXLWV4dHIuYXNwP2ZpbGVpZD0yMjAzMCZsYW5nPUVO&xsl=aHR0cDovL3NlbWFudGljcGFjZS5uZXQvWHNsdC9QZGYvWFJlZi1XRC1BVC1YTUwyUERGLnhzbA==&xsltparams=ZmlsZWlkPTIyMDMw

3) Therefore, while the role of industry in therapeutics and innovation is indisputable, must be encouraged and supported, the interest in the independence of prescribers is essential.

I disagree with this sentence, which for me corresponds more to the narrative of the pharmaceutical firms which wants to give themselves the best parts than facts. Prescrire showed that very few true innovative drugs have come to the markets in the last decades (Between 2002 and 2016, 52% of new drugs brought nothing new, 1% a real breakthrough, 6% an advantage, and 15% were not acceptable : Prescrire.

Feb 2017; 37 (400) : 133).

There are other arguments for this point that I presented in my PhD thesis (pages 24 o 31 : https://formindep.fr/these-de-paul-scheffer/) and people like Marcia Angell, former editor in chief of the NEJM, argues at the end of her book the pharmaceutical firms should not participate to the research and the education in medicine and shoud rather concentrate their action only making the drugs available everywhere they are needed (The Truth about Drug Companies).

At last, there are numerous calls for giving more public financial support to the public medical research, which seems to be much better than supporting firms, with some exemples given in this BMJ 2019 article https://www.bmj.com/content/367/bmj.l6576)

You may simply put it this way : Therefore the interest in the independence

of prescribers remains an essential goal, ten years after the publication of the Institute of Medicine report Conflicts of Interest in Medical

Research, Practice, and Education (you could here use this recent reference : https://ebm.bmj.com/content/early/2020/11/11/bmjebm-2020-111503)

4) However, their behavior remains rather positive towards

medical industries, as they still predominantly receive pharmaceutical representatives (80% of practitioners in France). [7, 8]

Here you could use this more recent reference instead, which gives the same number : https://journals.openedition.org/regulation/11272. I encourage the authors to read the whole paper to see if they are other elements that they would use for their article.

5) No study exploring the determining factors of such a contradiction has been found in France.

I don’t agree totally with this sentence because there is at least one work adressing this issue, although not specifically, whih is this article by the health anthropologist Anne Vega : https://www.cairn.info/revue-sciences-sociales-et-sante-2012-3-page-71.htm

I think to these two extracts of this document :

Enfin, les représentants des firmes sont aux côtés des étudiants dès la faculté [20][20]L’ensemble de ces constats (développés dans Vega, 2011b) sont à…, ce qui les habitue ensuite à une présence jugée naturelle, peu « chronophage » et/ou aidante

Si les enquêtés font plutôt confiance à l’expertise des médecins spécialistes, c’est par manque de moyens de contrôle sur leurs prescriptions, mais aussi parce qu’ils accordent une valeur plus faible à leurs propres savoirs et à leurs expériences généralistes, le plus souvent jugés peu scientifiques [28][28]Souligné aussi par Bloy (2008).. Cette dévalorisation contribue également à limiter le partage de décisions avec les confrères généralistes, expliquant des difficultés observées à centraliser les dossiers de patients. Elle expliquerait surtout des recours privilégiés, voire exclusifs, aux avis et aux médicaments prescrits par les médecins spécialistes, ou vantés comme tels par les représentants des firmes pharmaceutiques. Les savoirs spécialisés ou experts restent en effet des gages d’assurance et de prestige, voire les seules références, chez de jeunes médecins. Ainsi, des enquêtés ont le sentiment de « ne rien savoir » au sortir des facultés et sont constamment à la recherche de « recettes ». Non seulement ils reproduisent des prescriptions spécialisées, mais ils ajoutent à leur propre panel de médicaments des produits spécialisés.

According to this, a formulation like the following one may be more accurate :

One qualitative study gave some elements of comprehension of the the French GPs’ interactions with pharmaceutical sales representatives, like …, but no one tried to understand the contradiction why some general practitioners do receive the

visits of pharmaceutical representatives while they convey a rather negative opinion towards

pharmaceutical industries.

6) Several GPs in our sample have mentioned that pharmaceutical representatives and their bringing gifts are now less numerous.

Jerôme Greffion arrived at the same finding, you could mention it : https://journals.openedition.org/regulation/11272

7) You mention that among the interviewed GPs, one decided not to receive the reps anymore. Like Côté and Turgeon say (https://www.pedagogie-medicale.org/articles/pmed/abs/2002/02/pmed20023p81/pmed20023p81.html ), the deviant opinions are always very interesting to notice. Could he give some other reasons not mentioned by the others ?

8) About Limitations :

One limitation that could be added is the fact that the study, by its design, would not be able to collect elements of understanding 1) that would not be known by the interviewed doctors, for instance the fact that in France GP are quite isolated compared to the Dutch ones (see https://www.cairn.info/singuliers-generalistes--9782810900213-page-117.htm), which would reinforce the feeling of young doctors not to know and to rely on the information given by the reps. An other element conducting to the same result could be the fact that French medical students integrate the idea that they don’t have the right not to know during their training (see https://formindep.fr/these-de-paul-scheffer/ pages 177 to 181 and https://journals.openedition.org/ethiquepublique/1818), or 2) that they don’t want to speak about, linked to the fact that this topic is rather taboo.

9) This hypothesis has an external validity since some universities in the USA banned detailing visits

That is not exactly correct :

A growing number of US hospitals and academic medical centers (AMCs) have guidelines that limit interactions between physicians and pharmaceutical sales representatives. 1 No AMC has completely banned detailing, or the promoting of drugs directly to physicians by pharmaceutical salespeople. However, some AMCs have limited detailing visits—for instance, by restricting visiting hours or locations. Another strategy employed by AMCs is to ban physicians from receiving gifts or sample products from sales representatives (https://www.healthaffairs.org/doi/10.1377/hlthaff.2013.0939)

10) In 2017, Scheffer et Al. ranked French medical universities and teaching

hospitals according to their management policy for interest conflicts.

The study about teaching hospitals was published in 2019, not 2017.

11) The authors states that French universities are behind a lot of countries, including the USA, despite French students willingness to learn [23-26]. Indeed, the American Medical Student Association (AMSA) has been publishing a yearly ranking of American universities since 2007, with much better results.

This is not true anymore today, if we would use the AMSA score card type, all French medical schools which voted the Deans’ Conference of medicine ethical charter of 2017 would receive a A.

A formulation like this would be more accurate according to me :

Both studies showed that this topic was not adequately adressed by medical authorities yet, in spite of the ten years that have passed since the benfluorex scandal which led to a profound crisis in the French health sector. (suggested references : Mullard A (2011) Mediator scandal rocks French medical community. Lancet, 377: 890–892. pmid:21409784. Benkimoun P (2011) New law introduces tougher rules on drug regulation in France. BMJ 343: d8309. pmid:22194407. Fournier A, Zureik M. (2012) Estimate of deaths due to valvular insufficiency attributable to the use of benfluorex in France. Pharmacoepidemiol Drug Safety, 21: 343–351. Loi n° 2016–41 du 26 janvier 2016 de modernisation du système de santé français ). Some changes in the initial training could be fostered including our findings and precedent suggestions (https://www.pedagogie-medicale.org/articles/pmed/abs/2016/04/pmed170001/pmed170001.html)

Reviewer #3: The authors conducted a qualitative, phenomenological study to understand better why GPs in France continue to see pharmaceutical representatives while maintaining a critical view of the pharmaceutical industry. This is an important topic, but is one that has been previously explored (Chimonas et al, 2009, Fischer et al, 2008, Prosser 2008).

One of the key markers of rigour in qualitative research is “congruency” – that is, that the research aims and question are grounded in the theoretical orientation of the researchers (ontology, epistemology, e.g. socially constructed knowledge), which is consistent with the methodology (e.g. interpretive phenomenology) and methods (e.g. interviews) selected. Within this project, there are frequently mismatches that threaten the congruency of this work and introduce threats to the quality of the research.

Specifically, the researchers point to different disciplinary perspectives (“psychology, sociology and general practice”) as their theoretical grounding, but do not specify what these are or, how they worked together.

The researchers in the methods section, state that they used “interpretive phenomenology” but it is unclear in which tradition (citations are missing) and the tenets of this approach are not explicitly stated.

The methods chosen for sampling, interviewing and data analysis do not seem to be related to interpretive phenomenology (which emphasises narratives, observations, and action over reflection). The relationship between the methods, the methodology and the theoretical perspectives needs to at minimum, be explicit.

Finally, the researchers continue to mention “interpretive bias” in the methods and discussion and ways to mitigate this, however, the concept of “bias” is inconsistent with a qualitative, interpretive approach. Instead, the researchers should apply criteria for rigour that reflect research methods that emphasise the researcher’s role in data collection and analysis and that knowledge is socially constructed.

It appears the sample is a convenience rather than purposive sample. Sampling could be strengthened if individuals were recruited based on theoretically informed (rather than merely demographic) criteria. The researchers mention data saturation as a criterion for sampling; however, they do not demonstrate how they knew saturation was achieved and there does not seem to be sufficient presentation of the range of this phenomenon to suggest that they sampled the full range of this experience. This is not a critique of the sample size, but rather, that qualitative researchers must demonstrate to the reader that they sought contrasting perspectives and individuals with diverse experience until no new information arose.

The researchers stated that this is an ‘exploratory’ analysis, but also seem to try and theorize this phenomenon. Thus, the aims of this work need clarification. If this is more of a theoretical analysis (around hidden curriculum, or cognitive dissonance, e.g. which I think would be of high value), then the theoretical perspectives informing this work and past work on this topic should be brought up front to really highlight what this work adds. At the moment, it is unclear which literature this work is in conversation with and the discussion really contains a lot of useful analysis, but fails to put it into context.

At the moment, the analysis remains description, but could be greatly strengthened by incorporating the material in the discussion. The researchers should clarify whether they are applying particular theoretical perspectives, or are conducting a grounded, interpretive analysis (per the stated interpretive phenomenology), and then work to refine the results in either direction.

This work is important and I encourage the researchers to continue this work. However, I think it may need to be reworked from the start to really clarify the aim, the theory/methods package, and to re-analyse the findings in this vein.

This work is important and I encourage the researchers to continue this work. However, I think it may need to be reworked from the start to really clarify the aim, the theory/methods package, and to re-analyse the findings in this vein.

6. PLOS authors have the option to publish the peer review history of their article (what does this mean?). If published, this will include your full peer review and any attached files.

Reviewer #1: **Yes: **Joel Lexchin

Reviewer #2: **Yes: **Paul Scheffer

Reviewer #3: **Yes: **Quinn Grundy, University of Toronto

---

## [Author Response · Author response to Decision Letter 0]

31 May 2021

Answers to reviewers:

Dear Barbara Mintzes, Academic Editor,

Dear Joel Lexchin, Paul Scheffer and Quinn Grundy, reviewers,

Thank you for giving us the opportunity to upgrade our manuscript. We have considered all your comments with interest. Please find below our point-by-point answer.

Academic Editor reports:

This article addresses an important issue, the relationship between primary care physicians and pharmaceutical sales representatives. The discussion is thoughtful and the focus on cognitive dissonance resonates with other related research. Currently, inadequate information is provided on methods (selection of participants, study conduct, and approach to analysis) to adequately assess the article's contribution to the existing literature on this issue. 

Thank you. The methods section has been clarified (see below).

Additionally, the added French context needs to be considered, as is outlined below, given the extra restrictions on sales visits in France as compared with most countries. This was a key theme emerging in a previous qualitative study (Reynolds et al. 2018 - full ref below).

Thank you for the additional reference that was added to ours. The French context is now described in detail in the introduction (lines 137-161). In the discussion – comparison with other studies, a sentence has been added : “In contrast to a study published in 2018,[44] physicians talked relatively little about regulations and did not talk about drug-specific safety scandals such as those regarding rofecoxib (Vioxx) and benfluorex (Mediator). This difference is likely due to the interview guide, which sought to explore the reasons for the discordance between attitude and behavior and did not allow these themes to emerge. Likewise, physicians did not talk about the sales visit charter, suggesting that it remains relatively unknown more than 10 years after its adoption. Instead, they described a decrease in the frequency of medical visitors and in the size of the gifts offered to them. This decrease in the number and volume of gifts is confirmed by the analysis of the national “transparence-santé” database.[43] This decrease was used by doctors as an argument to trivialize the impact of sales rep visits on their practice. Although we can welcome the decrease in gifts, this trivialization may contribute to ongoing GP ambivalence towards sales visits.”

As is noted below, given that three case scenarios with very different questioning afterwards were included in this study, reporting on how doctors responded to these scenarios and links to their views on sales visits need to be made more explicit. This is likely a valuable component of the study data and it is unclear why it was omitted from results reporting. 

The scenarios served to gradually bring up the issue of sales visits without triggering the reluctance and social desirability bias observed in the two preliminary interviews. The questions that followed the three scenarios were similar. The number of scenarios only served to renew the discussion if necessary, by moving on to the next scenario. […] The three scenarios did not each provide a different pattern of responses, which was expected since the research question was not at all about the drugs discussed in the scenarios, but about the doctor-sales rep relationship. This clarification was made in the introduction (line 259) and the results.

Additionally, I had concerns about adequacy of assessment of data saturation with 3 different scenarios & questioning and 10 interviews in total. You may want to reconsider whether additional data are needed as well as a re-analysis. 

Thank you for this comment. Each physician responded to all three scenarios, which explains how we reached data saturation with 10 interviews. We have clarified this point in the methods section.

Here are my detailed recommendations: Clarification of methods and ensuring that there is an explicit link between how methods are described and how the analysis is conducted and results reported and discussed. Dr Grundy describes this in detail in her comments, specifically in relation to the theoretical framing of the study and both the need for a more explicit description in the methods, with citations, and a clearer link between this framework and how results are analyzed and presented. 

Thank you. Each of Dr. Grundy's comments has been carefully considered. I have not copied every comment here so as not to lengthen the document, please see below.

In addition the approach to selection of study participants (convenience, snowball, purposive?)

“A convenience sampling method was used to select people with a range of experience in meeting sales reps. No specific inclusion or non-inclusion criteria were imposed since the research team considered that any GP could contribute new data to explain the phenomenon studied. The selection of practitioners was based on variation criteria for the following characteristics: …” 

 how many were contacted and agreed or refused needs to be reported in greater detail?

Done: “Ten GPs refused to participate because of lack of time.” 

 Were there specific inclusion criteria (for example that the interviewed GPs saw reps regularly)?

Added: “No specific inclusion or non-inclusion criteria were imposed since the research team considered that any GP could contribute new data to explain the phenomenon studied.”

If the aim was to explore effects of specific factors linked to demographics, for example gender differences, it’s unclear why only 2 female GPs were included, given a 44% female GP workforce in France, and also whether differences by demographics (age, gender) and number of years in practice or type of practice were considered as factors in assessment of whether saturation had been reached. 

The aim was not to explore the effects of specific factors linked to demographics but to gather a wide range of GPs without missing any demographic criteria. We added in the methods section, line 220 “The purpose of a phenomenological approach is mainly to shed light on the essence of a person’s experience in relation to a specific phenomenon, in this instance meeting sales reps despite holding a negative opinion of them. In a phenomenological study, sample sizes are typically small because data saturation is not essential if the objective is to understand the meaning individuals give to their own lived experience. (Van Manen et al, 2016). 

It is also unclear when the interviews were carried out. This needs to be stated directly.

This is stated at the beginning of the results section: “The material is composed of 10 interviews carried out from April to December 2014, of an average duration of 50 minutes (median of 46 minutes), for a total duration of 8 hours and 22 minutes, representing 118 pages of transcriptions”.

How the case scenarios informed the discussion and interpretation of results needs to be better reported. A study strength is the piloting of the approach used to questioning, and the introduction of revisions to avoid a social desirability bias and/or defensive responses to questioning. As these are two separate However, it is unclear how the three case scenarios were incorporated into this study, especially given the fundamental differences in approach to questioning following each scenario. The description of these scenarios in the article remains general and does not highlight these differences, nor include even a brief description of the drug and context per scenario.

I apologize for the lack of precision in the way the interview guide was described. In fact, the questions following each scenario were similar. Once again, the number of scenarios was only meant to renew the discussion when necessary, within a different context. The interview guide (supplementary materials) has been modified to be more explicit and the description of the scenarios has been enhanced : 

“The scenarios served to gradually bring up the issue of sales rep visits without causing the reluctance and social desirability bias observed in the two preliminary interviews. The questions that followed the three scenarios were similar (“What sources of information do you regularly use?” – “Can you tell me more about sales reps?”). The next questions aimed at progressively focusing on the GP/rep relation (“Have you ever considered no longer meeting reps?” – “I am particularly interested in the apparent contradiction between what you tell me” ... “and, at the same time,” ... “How did you come to this opinion?”). Some questions were not directly related to sales reps in order to avoid defensive reactions observed in the preliminary interviews. The second and the third scenarios only served to renew the discussion when necessary by moving on to the next scenario. The first scenario concerned the prescription of sitagliptin to a frail patient, the second the switch to oral anticoagulation in an elderly and poly-medicated patient and the third the prescription of rivaroxaban to a young patient with paraphlebitis, following the advice of a colleague.” 

 Were the specific drugs selected based on lack of evidence of therapeutic advantages or cost-effectiveness ratings? 

Yes. The specific drugs were selected based on lack of evidence of therapeutic advantages or cost effectiveness because previous studies have shown that it is in a situation of doubt that a practitioner is the most sensitive to the influence of pharmaceutical representatives.

How did doctors respond to the specific scenarios? 

The aim of the scenarios was not to collect different answers and analyze them separately but to allow the discussion to go deeper during the interview.

And were there any clear links between their responses and attitudes and experiences towards sales representatives? 

We did not question GPs further about what they thought about the scenarios, the discussion quickly turned to their relationship with pharma sales reps.

It is also unclear how these scenarios functioned in practice – did all of the doctors spontaneously mention sales representatives in response to these scenarios on a new drug prescription, or did they mention other factors more often and researchers introduced the link to sales representatives?

All GPs but one spontaneously mentioned sales reps following the scenarios. When the physician did not address the visit spontaneously, the interviewer addressed it specifically (added in the results section).

 Additionally, why were three separate scenarios used with different associated questions – some more directly related to sales representatives than others - and how often was each one used? 

Some questions were not directly related to sales reps in order to avoid defensive reactions observed in the preliminary interviews (added in the methods section).

All questions were used for all GPs (added in the interview guide).

When data saturation was being considered, how did the use of multiple scenarios affect judgments of thematic saturation?

Since all scenarios were used for all interviewed GPs, the use of multiple scenarios did not affect judgments of thematic saturation.

The study could be much better informed by the unique position of GPs in France in relation to reps as compared with most other countries, in that free medication samples and gifts of food and drink within the office are not allowed, and an attempt has been made to ensure information quality through public oversight of a certification procedure.

Added in the introduction: “The role of pharmaceutical sales representatives (sales reps) is regulated by the government through a Sales Visit Charter and a certification procedure but despite more stringent regulatory requirements than in North America, the information provided by sales reps remains poor.[5,6] The Sales Visit Charter Promotes good practices such as the presentation of approved product information including side effects and contraindications. Despite strict regulations (health authorities can impose fines of up to €10,000 or 10% of a product’s annual sales revenues) [7], the Sales Visit Charter is rarely respected.[5,6] Pharma sales reps are not allowed to follow continuing medical education courses. A national fund for continuing medical education finances free courses for GPs and compensates for the loss of three days’ worth of medical fees while training. This non-biased continuing medical education is mandatory.[8] Initial and continuing medical education are supposed to be provided by impartial experts but these experts can be opinion leaders with financial ties to the industry.[9–12] Despite these measures, the pharmaceutical industry still plays an active role in research and in general practitioner (GP) training from the preclinical years to continuing medical education.[11] The French National Health Authority, “Haute Autorité de Santé” (HAS), encourages professionals and healthcare facilities to address the issue of sales visits and provides tools such as a French translation of the World Health Organization’s guide “Understanding and responding to pharmaceutical promotion: a practical guide”. HAS also provides guidance to health care facilities on how to regulate sales visits.[13,14] […] Free drug samples and gifts of food and drink in the office are not allowed in France, but GPs still receive gifts and invitations to restaurants since the law allows for exceptions.” 

 Additionally, La Revue Prescrire used a network of GPs to analyse the content of sales visits in France over a 15-year period. (La Revue Prescrire 2006; 26(272):383-389) The context and French policies such as the sales visit charter would be expected to have some influence. How was the approach to questioning and analysis informed by this context? 

Thank you for this reference to La Revue Prescrire, this work is very interesting and has been added to our references (in the introduction and the discussion). In our study, physicians did not talk about the sales visit charter, suggesting that it remains relatively unknown more than 10 years after it was adopted. Instead, they described a decrease in the frequency of sales rep visits and in the size of the gifts offered to them. This decrease in the number and volume of gifts is confirmed by the analysis of the national “transparence-santé” database.[43] This decrease was used by doctors as an argument to trivialize the impact of sales rep visits on their practice. Although we can welcome the decrease in gifts, this trivialization may contribute to ongoing GP ambivalence towards sales visits. (added in the discussion – comparison with other studies)

Also please include a description of the current approach to oversight. See for example: https://www.has-sante.fr/jcms/c_1099657/fr/visite-medicale-accompagner-les-professionnels-et-les-etablissements; Habibi et al. Journal of Law, Medicine and Ethics. 2016; 44: 602-613; Reynolds et al. Health Policy 2018; 122(3): 250-255; Mintzes et al. JGIM 2013; 28(10): 1368–1375. 

Done. (please see above - introduction) 

In the focus groups described in Reynolds et al., we also found that a strong influence from the Mediator scandal in how the French GPs discussed sales rep visits. This is likely to have been influenced by timing of the focus groups but an open question is whether in the current interview study there was any spontaneous discussion of safety concerns and/or of Mediator.

Interviewed GPs spoke little about safety concerns and/or of Mediator. This has been added to the results:

“Interviewed GPs raised few safety concerns about meeting reps: “A few years ago, it had become the only source of information. I mean that those who didn’t read ‘La Revue Prescrire’, they’d prescribe; they followed the information brought by the visitors. It’s dangerous to do that! But that’s changed; it’s changed a lot.” None of them talked about the sales visit charter or drug-specific safety scandals.”

And to the discussion: 

“In contrast to a study published in 2018,[44] physicians talked relatively little about regulations and did not talk about drug-specific safety scandals such as those regarding rofecoxib (Vioxx) and benfluorex (Mediator). This difference is likely due to the interview guide, which sought to explore the reasons for the discordance between attitude and behavior and did not allow these themes to emerge. Likewise, physicians did not talk about the sales visit charter, suggesting that it remains relatively unknown more than 10 years after its adoption. Instead, they described a decrease in the frequency of medical visitors and in the size of the gifts offered to them. This decrease in the number and volume of gifts is confirmed by the analysis of the national “transparence-santé” database.[43] This decrease was used by doctors as an argument to trivialize the impact of sales rep visits on their practice. Although we can welcome the decrease in gifts, this trivialization may contribute to ongoing GP ambivalence towards sales visits.”

As noted by Dr Lexchin, editing of the manuscript by an English-language editor would be helpful. There are a few repeated “false friends”. For example ‘laboratoires’ translates to pharmaceutical companies, not to labs. In English a laboratory or lab is a facility where scientific experiments of all types take place. A ‘lab scientist’ is someone using usually very expensive specialised equipment (in their lab) to carry out scientific experiments.

Thank you for these explanations. The manuscript has been thoroughly copy-edited by a native English speaker (Yvonne van der Does from the Office of International Scientific Visibility, Université Côte d'Azur)

Thank you, modifications have been brought thanks to these templates (font, figure captions etc.).

2. Please consider changing the title so as to meet our title format requirement (https://journals.plos.org/plosone/s/submission-guidelines). In particular, the title should be "Specific, descriptive, concise, and comprehensible to readers outside the field" and in this case it could be preferable to avoid undefined acronyms

Done.

3. We suggest you thoroughly copyedit your manuscript for language usage, spelling, and grammar. If you do not know anyone who can help you do this, you may wish to consider employing a professional scientific editing service. Upon resubmission, please provide the following: a) The name of the colleague or the details of the professional service that edited your manuscript b) A copy of your manuscript showing your changes by either highlighting them or using track changes (uploaded as a *supporting information* file) c) A clean copy of the edited manuscript (uploaded as the new *manuscript* file)

Done.

Datasets have been de-identified and are available in supporting information file S1.

5. Thank you for stating the following financial disclosure: "NO - The funders had no role in study design, data collection and analysis, decision to publish, or preparation of the manuscript." At this time, please address the following queries:

Please clarify the sources of funding (financial or material support) for your study. List the grants or organizations that supported your study, including funding received from your institution.

State what role the funders took in the study. If the funders had no role in your study, please state: “The funders had no role in study design, data collection and analysis, decision to publish, or preparation of the manuscript.”

If any authors received a salary from any of your funders, please state which authors and which funders.

If you did not receive any funding for this study, please state: “The authors received no specific funding for this work.”

Thank you for raising this point. Indeed, the authors received no specific funding for this work. The financial disclosure has been modified directly in the editorial manager system.

Done. The authors received no specific funding for this work.

Done.

7. Please include captions for your Supporting Information files at the end of your manuscript, and update any in-text citations to match accordingly. Please see our Supporting Information guidelines for more information: http://journals.plos.org/plosone/s/supporting-information

Done.

This paper does not contain any figures.

Reviewers' comments:

Reviewer #1: This is a qualitative study looking at why French GPs see sales representatives even though they generally have negative attitudes about the pharmaceutical industry. This type of inquiry is useful in formulating strategies about how to minimize the contact between these two groups and as such the authors should put forward policies towards that end based on their research.

We added in the discussion (line 560): “Studies showed that this topic was still not adequately addressed by medical authorities, even though ten years have passed since the benfluorex scandal which led to a profound crisis in the French health sector.[51–53] Some changes in the initial training could be introduced including our findings and previous suggestions: prohibiting professors from receiving gifts from sales reps, nudging them towards greater independence, providing a minimum curriculum for all medical students such as those already existing in France or elsewhere.[50,54–57] Continuing medical education could also be enhanced. GPs effectively felt the lack of free, non-biased and easy-to-access educational options that match their organizational constraints. French continuing medical education still suffers from a lack of independence and transparency in congresses and clinical practice guidelines despite the legal obligation for experts to disclose competing interests.[42,58] The development of convenient, non-biased, and individual continuing medical education is therefore a public health necessity.”

1. The English in many places is awkward and the entire manuscript should be copy edited by someone who has English as a first language.

Done. (see above)

2. It would help in commenting on the manuscript if the lines were numbered.

Done.

3. In describing how interactions with sales representatives influence prescribing the authors should cite some of the literature coming out of the use of the Open Payments database.

Done (line 95): “Financial relationships between the leaders of influential professional US medical associations and industry are extensive: 72% of them have financial ties to industry.[2]”.

4. Page 4: I think that it's highly likely that a portion of the medical profession is familiar with the terms cognitive dissonance and hidden curriculum and this sentence should be modified so as not to imply that it applies to the entire profession.

Thank you for raising that point. The sentence has been modified (line 88): “The GP/sales representative relationship is complex and involves psychological mechanisms that the medical profession tends to ignore such as cognitive dissonance and a hidden curriculum”

5. Page 4: The part of the sentence "complete the latter in a liberal field" is not clear.

We clarified this point as follows: « Sales representative visits act as a fast and easily accessible educational service that fills a need expressed by practitioners in France.»

6. Page 5: The authors should give a bit of background about general practice in France - how are doctors paid, do they generally practice alone or in groups, is there any funding for continuing medical education, do they also work in hospitals, etc.

We added the following details (lines 123-137): “A national fund for continuing medical education finances free courses for GPs and compensates for the loss of three days’ worth of medical fees while training. This non-biased continuing medical education is mandatory.[8] […] French primary care is mainly organized as a private, independent service, although financed by the public health insurance. Like most primary care providers in France, GPs mainly work in private practice and are paid on a fee-for-service basis. Group practice is now dominant among GPs and 81% of GPs younger than 50 declare that they work in a group.[15] Since the 2000s, teamwork in multidisciplinary primary care centers has been developed, mainly to maintain primary care services in low-income areas”

7. Page 5: USA should be spelled out in full the first time it is used.

Done

8. Page 5: The regulation of the activities of pharmaceutical representatives should be briefly described along with any penalties for violating the regulations.

Done: “The role of pharmaceutical sales representatives (sales reps) is regulated by the government through a Sales Visit Charter and a certification procedure but despite more stringent regulatory requirements than in North America, the information provided by sales reps remains poor.[5,6] The Sales Visit Charter Promotes good practices such as the presentation of approved product information including side effects and contraindications. Despite strict regulations (health authorities can impose fines of up to €10,000 or 10% of a product’s annual sales revenues) [7], the Sales Visit Charter is rarely respected.[5,6] Pharma sales reps are not allowed to follow continuing medical education courses”

9. Page 5: In what ways does the industry take part in the training of GPs?

We added the following details (line 111): “Initial and continuing medical education are supposed to be provided by impartial experts but these experts can be opinion leaders with financial ties to the industry.[9–12] Despite these measures, the pharmaceutical industry still plays an active role in research and in general practitioner (GP) training from the preclinical years to continuing medical education.[11]”

Moreover, “after the drug database Vidal, sales rep visits were the second source of information for practitioners.” (line 135).

To finish with, many congresses and reviews are funded by the industry.

10. Page 6: The authors should cite some of the literature showing that GPs believe in their own ability to resist being influenced by representatives but not necessarily in the ability of other GPs.

Done

11. Page 6: There should be a reference to the literature describing the phenomenological approach.

Done (Smith J. A, Flowers P, Larkin M. Interpretative phenomenological analysis: Theory, method and research. London: Sage; 2009. The methodological paradigm for this study was also specified.)

12. Page 7: The existing literature used to generate the interview guide should be cited.

Thank you for raising that point! It did not rely on a conceptual framework but only on the literature cited in the introduction. We changed to: “The interview guide was developed from existing literature[23] (identifying professional situations of uncertainty and reasons for meeting sales reps) and from two preliminary interviews.”.

13. Page 7: What GP details are the authors referring to? The city where the study took place is mentioned later but should be mentioned here. How did the authors determine if the GPs saw sales representatives? Did all GPs invited agree and if not how many refused?

GPs details has been replaced by “GPs’ phone numbers”.

The mention of the city has been moved here.

This part has been clarified as follows: “Participants were asked about these characteristics at the end of the interview” ; “Ten GPs refused to participate because of a lack of time.”.

14. Page 9: I'm not sure how to interpret the final sentence in the Methods section.

“The investigator’s motives and interests for the research subject have been questioned and analyzed beforehand within the research team.” This assertion relates to the COREQ criteria “What characteristics were reported about the interviewer/facilitator? e.g. Bias, assumptions, reasons and interests in the research topic”. This sentence has been reworded as follows: “The main investigator examined his own assumptions, and his reasons and interests for studying the research topic by brainstorming with the research team before the research started”

15. Page 9: How does peer group continuing education function in France?

Quality circles or peer groups are structured small groups of 6 to 12 health care professionals who meet regularly to reflect on and improve their practice. They also serve to develop professional networks. They are very popular in France, where participation can give rise to an indemnity when the group is accredited.[31]. Precisions have been added to the paper (line 211)

16. Page 10 (Table 1): What does "setting as a team" mean in the French context?

We meant « Group practice ». It has been clarified. 

17. Page 10 (last paragraph): The first sentence is a repeat of what was said in the Introduction and Methods and should be deleted.

Done

18. Page 11: What does "counter-phobic effect" mean?

Clarified as follows: The participants attributed to rep visits a reassurance effect regarding their knowledge, and a “starter” effect as the first link of the information chain: meeting a sales rep was said to trigger more extensive research.

19. Page 12: What does “training session” mean?

We meant “course”. This part has been clarified.

20. Page 15: The part of this paragraph starting "Thus, these data..." is a conclusion and belongs in that section.

It has been moved.

21. Page 17: What “tendency” are the authors talking about?

Clarified as follows: “Several French GPs said that sales reps were fewer and brought gifts less often. Other observational studies arrived at the same conclusion.[24,43].” 

22. Page 19: AMSA stopped publishing rankings in 2016.

Thank you. The sentence has been deleted.

Reviewer #2: 

Dear authors,

Thank you first for this interesting and useful work. I used the Côté-Turgeon assessment grid (https://www.pedagogie-medicale.org/articles/pmed/abs/2002/02/pmed20023p81/pmed20023p81.html )

to evaluate your article and all the points listed got a yes, there are so only minor corrections to do according to me.

1) I would rephrase this extract like following:

Many studies show a correlation between receiving a gift from pharma reps or meeting them

and an amount of bigger, more expensive and sometimes less rational prescriptions [give here the corresponding references, this one should belong to them I think: https://bmjopen.bmj.com/content/7/9/e016408 ]. A recent study published in the BMJ confirmed this for French GPs through indicators such as prescription of antibiotics, benzodiazepines and vasodilators [4]

Done, thank you.

2) This situation is as much of a problem on the financial level (with costs that are indirectly born by the community via medication prices) as it is on the social, ethical and Public Health levels. [2]

You could add this more recent reference here: Public health and the interests of the pharmaceutical industry: how to guarantee the primacy of public health interests? European Parliamentary Assembly, 2015: http://semantic-pace.net/tools/pdf.aspx?doc=aHR0cDovL2Fzc2VtYmx5LmNvZS5pbnQvbncveG1sL1hSZWYvWDJILURXLWV4dHIuYXNwP2ZpbGVpZD0yMjAzMCZsYW5nPUVO&xsl=aHR0cDovL3NlbWFudGljcGFjZS5uZXQvWHNsdC9QZGYvWFJlZi1XRC1BVC1YTUwyUERGLnhzbA==&xsltparams=ZmlsZWlkPTIyMDMw

Done, thank you.

3) Therefore, while the role of industry in therapeutics and innovation is indisputable, must be encouraged and supported, the interest in the independence of prescribers is essential.

I disagree with this sentence, which for me corresponds more to the narrative of the pharmaceutical firms which wants to give themselves the best parts than facts. Prescrire showed that very few true innovative drugs have come to the markets in the last decades (Between 2002 and 2016, 52% of new drugs brought nothing new, 1% a real breakthrough, 6% an advantage, and 15% were not acceptable: Prescrire.Feb 2017; 37 (400): 133).

There are other arguments for this point that I presented in my PhD thesis (pages 24 o 31: https://formindep.fr/these-de-paul-scheffer/) and people like Marcia Angell, former editor in chief of the NEJM, argues at the end of her book the pharmaceutical firms should not participate to the research and the education in medicine and should rather concentrate their action only making the drugs available everywhere they are needed (The Truth about Drug Companies).

At last, there are numerous calls for giving more public financial support to the public medical research, which seems to be much better than supporting firms, with some examples given in this BMJ 2019 article https://www.bmj.com/content/367/bmj.l6576 )

You may simply put it this way: Therefore the interest in the independence of prescribers remains an essential goal, ten years after the publication of the Institute of Medicine report Conflicts of Interest in Medical Research, Practice, and Education (you could here use this recent reference: https://ebm.bmj.com/content/early/2020/11/11/bmjebm-2020-111503 )

Done, thank you.

4) However, their behavior remains rather positive towards medical industries, as they still predominantly receive pharmaceutical representatives (80% of practitioners in France). [7, 8]

Here you could use this more recent reference instead, which gives the same number: https://journals.openedition.org/regulation/11272 . I encourage the authors to read the whole paper to see if they are other elements that they would use for their article.

Done, thank you. Great paper!

5) No study exploring the determining factors of such a contradiction has been found in France.

I don’t agree totally with this sentence because there is at least one work addressing this issue, although not specifically, which is this article by the health anthropologist Anne Vega: https://www.cairn.info/revue-sciences-sociales-et-sante-2012-3-page-71.htmI think to these two extracts of this document:

Enfin, les représentants des firmes sont aux côtés des étudiants dès la faculté [20][20]L’ensemble de ces constats (développés dans Vega, 2011b) sont à…, ce qui les habitue ensuite à une présence jugée naturelle, peu « chronophage » et/ou aidante

Si les enquêtés font plutôt confiance à l’expertise des médecins spécialistes, c’est par manque de moyens de contrôle sur leurs prescriptions, mais aussi parce qu’ils accordent une valeur plus faible à leurs propres savoirs et à leurs expériences généralistes, le plus souvent jugés peu scientifiques [28][28]Souligné aussi par Bloy (2008).. Cette dévalorisation contribue également à limiter le partage de décisions avec les confrères généralistes, expliquant des difficultés observées à centraliser les dossiers de patients. Elle expliquerait surtout des recours privilégiés, voire exclusifs, aux avis et aux médicaments prescrits par les médecins spécialistes, ou vantés comme tels par les représentants des firmes pharmaceutiques. Les savoirs spécialisés ou experts restent en effet des gages d’assurance et de prestige, voire les seules références, chez de jeunes médecins. Ainsi, des enquêtés ont le sentiment de « ne rien savoir » au sortir des facultés et sont constamment à la recherche de « recettes ». Non seulement ils reproduisent des prescriptions spécialisées, mais ils ajoutent à leur propre panel de médicaments des produits spécialisés.

According to this, a formulation like the following one may be more accurate:

One qualitative study gave some elements of comprehension of the French GPs’ interactions with pharmaceutical sales representatives, like …, but no one tried to understand the contradiction why some general practitioners do receive the visits of pharmaceutical representatives while they convey a rather negative opinion towards pharmaceutical industries.

Done

6) Several GPs in our sample have mentioned that pharmaceutical representatives and their bringing gifts are now less numerous.

Jerôme Greffion arrived at the same finding, you could mention it: https://journals.openedition.org/regulation/11272

Done

7) You mention that among the interviewed GPs, one decided not to receive the reps anymore. Like Côté and Turgeon say (https://www.pedagogie-medicale.org/articles/pmed/abs/2002/02/pmed20023p81/pmed20023p81.html ), the deviant opinions are always very interesting to notice. Could he give some other reasons not mentioned by the others ?

Interestingly, this GP didn’t express a very “deviant opinion” with respect to the aim of our paper. The interview focused on his previous experience with meeting reps and the transcripts were analyzed in the same way as for the other participants.

8) About Limitations:

One limitation that could be added is the fact that the study, by its design, would not be able to collect elements of understanding 1) that would not be known by the interviewed doctors, for instance the fact that in France GP are quite isolated compared to the Dutch ones (see https://www.cairn.info/singuliers-generalistes--9782810900213-page-117.htm ), which would reinforce the feeling of young doctors not to know and to rely on the information given by the reps. Another element conducting to the same result could be the fact that French medical students integrate the idea that they don’t have the right not to know during their training (see https://formindep.fr/these-de-paul-scheffer/ pages 177 to 181 and https://journals.openedition.org/ethiquepublique/1818), or 2) that they don’t want to speak about, linked to the fact that this topic is rather taboo.

Added: “The study’s design did not allow the authors to collect elements of understanding unknown to the interviewed doctors, for instance the fact that in France GPs are quite isolated compared to Dutch ones.[45] This could have reinforced the feeling young doctors had of lacking knowledge and their need to rely on the information provided by the reps.”

9) This hypothesis has an external validity since some universities in the USA banned detailing visits

That is not exactly correct: A growing number of US hospitals and academic medical centers (AMCs) have guidelines that limit interactions between physicians and pharmaceutical sales representatives. 1 No AMC has completely banned detailing, or the promoting of drugs directly to physicians by pharmaceutical salespeople. However, some AMCs have limited detailing visits—for instance, by restricting visiting hours or locations. Another strategy employed by AMCs is to ban physicians from receiving gifts or sample products from sales representatives (https://www.healthaffairs.org/doi/10.1377/hlthaff.2013.0939 )

Done: “These observations lead us to consider the hypothesis that meetings with sales reps during initial training shape the GP-rep relationship which is then maintained over time by the three broad explanations described above. It has been demonstrated that the prohibition of gifts or samples has a positive influence on prescriptions well over several years after students leave the university.[46,47] These data bring external validity to our hypothesis.”

10) In 2017, Scheffer et Al. ranked French medical universities and teaching hospitals according to their management policy for interest conflicts. The study about teaching hospitals was published in 2019, not 2017.

Modified.

11) The authors state that French universities are behind a lot of countries, including the USA, despite French students willingness to learn [23-26]. Indeed, the American Medical Student Association (AMSA) has been publishing a yearly ranking of American universities since 2007, with much better results.

This is not true anymore today, if we would use the AMSA score card type, all French medical schools which voted the Deans’ Conference of medicine ethical charter of 2017 would receive an A. A formulation like this would be more accurate according to me: “Both studies showed that this topic was not adequately addressed by medical authorities yet, in spite of the ten years that have passed since the benfluorex scandal which led to a profound crisis in the French health sector. (suggested references: Mullard A (2011) Mediator scandal rocks French medical community. Lancet, 377: 890–892. pmid:21409784. Benkimoun P (2011) New law introduces tougher rules on drug regulation in France. BMJ 343: d8309. pmid:22194407. Fournier A, Zureik M. (2012) Estimate of deaths due to valvular insufficiency attributable to the use of benfluorex in France. Pharmaco Perlis, Roy H., et Clifford S. Perlis. « Physician Payments from Industry Are Associated with Greater Medicare Part D Prescribing Costs ». PLoS ONE 11, no 5 (16 mai 2016). https://doi.org/10.1371/journal.pone.0155474. epidemiol Drug Safety, 21: 343–351. Loi n° 2016–41 du 26 janvier 2016 de modernisation du système de santé français ). Some changes in the initial training could be fostered including our findings and precedent suggestions (https://www.pedagogie-medicale.org/articles/pmed/abs/2016/04/pmed170001/pmed170001.html )”

Thank you for this suggestion. This part has been modified as follows: “Studies showed that this topic was still not adequately addressed by medical authorities, even though ten years have passed since the benfluorex scandal which led to a profound crisis in the French health sector.[51–53] Some changes in the initial training could be introduced including our findings and previous suggestions: prohibiting professors from receiving gifts from sales reps, nudging them towards greater independence, providing a minimum curriculum for all medical students such as those already existing in France or elsewhere.[50,54–57] Continuing medical education could also be enhanced. GPs effectively felt the lack of free, non-biased and easy-to-access educational options that match their organizational constraints. French continuing medical education still suffers from a lack of independence and transparency in congresses and clinical practice guidelines despite the legal obligation for experts to disclose competing interests.[42,58] The development of convenient, non-biased, and individual continuing medical education is therefore a public health necessity”

Reviewer #3: 

The authors conducted a qualitative, phenomenological study to understand better why GPs in France continue to see pharmaceutical representatives while maintaining a critical view of the pharmaceutical industry. This is an important topic, but is one that has been previously explored (Chimonas et al, 2009, Fischer et al, 2008, Prosser 2008).

Thank you for raising that point. Both of these papers tried to understand the reasons why GPs meet pharmaceutical representatives and their results are complementary with ours. The originality of our paper is to focus on understanding the discrepancy between GP opinions and behaviors. Both of these papers are discussed in the discussion of the revised manuscript.

One of the key markers of rigor in qualitative research is “congruency” – that is, that the research aims and question are grounded in the theoretical orientation of the researchers (ontology, epistemology, e.g. socially constructed knowledge), which is consistent with the methodology (e.g. interpretive phenomenology) and methods (e.g. interviews) selected. Within this project, there are frequently mismatches that threaten the congruency of this work and introduce threats to the quality of the research. Specifically, the researchers point to different disciplinary perspectives (“psychology, sociology and general practice”) as their theoretical grounding, but do not specify what these are or, how they worked together.

(i) Concerning congruency, we agree that it was not clear enough and we made the following modifications: 

- INTRODUCTION SECTION

“This study explores GP experiences to better understand their ambivalence”

- METHODS SECTION

“An exploratory qualitative study was conducted based on a constructivist paradigm. An interpretative phenomenological approach was used to explore individual professional practices and to model the phenomenon through the in-depth analysis of singular experiences, of the meaning that participants gave to this experience and of underlying psychological mechanisms.[25].”

“The interview guide was developed from existing literature[23] (identifying professional situations of uncertainty and reasons for meeting sales reps) and from two preliminary interviews.”

“A convenience sampling method was used to select people with a range of experience in meeting sales reps. No specific inclusion or non-inclusion criteria were imposed since the research team considered that any GP could contribute new data to explain the phenomenon studied. The selection of practitioners was based on variation criteria for the following characteristics: […] These characteristics were expected to lead to a diverse sample of experiences and were based on the literature describing how the GP/rep relationship depends on social norms, age, patient list size and the type of continuing medical education.[4,23].”

“Transcripts were analyzed one interview at a time to explore individual perspectives. Texts were commented, then themes were coded, organized and finally formulated to identify common patterns shared by the different experiences.[25,27–29] Results were interpreted by a multi-field team composed of three GPs (AB, IP, TB), a sociologist (IF), and a social psychologist (IM). 

“Data saturation was discussed and agreed upon by the research team when the sample was considered as gathering a wide range of experiences about relations to sales reps and no new theme emerged from the material analyzed. Theoretical validation was obtained by comparing the results with existing scientific data (see discussion)”

(ii) Concerning the diversity of disciplines collaborating in the project: this was not so much to provide a theoretical grounding but to anticipate the emerging elements of comprehension. Preparation with the workgroup was designed to better identify the emerging concepts and their theoretical groundings. We anticipated that concepts would relate to cognition but also possibly to social interactions: that’s why we gathered an interdisciplinary team. The manuscript modifications mentioned above add details about our grounding which related more to psychology and behavior, and justifies our phenomenological approach and the interview guide designed to explore personal experiences.

We added: “The heterogeneity of the team’s skills aimed at identifying themes more broadly and enlarging the scope of the analysis.”

The researchers in the methods section, state that they used “interpretive phenomenology” but it is unclear in which tradition (citations are missing) and the tenets of this approach are not explicitly stated.

Thank you again for this opportunity for clarification, modifications have been made (cf previous comment)

The methods chosen for sampling, interviewing and data analysis do not seem to be related to interpretive phenomenology (which emphasizes narratives, observations, and action over reflection). The relationship between the methods, the methodology and the theoretical perspectives needs to at minimum, be explicit.

Thank you again for this opportunity for clarification, modifications have been made (cf. preceding comment)

Finally, the researchers continue to mention “interpretive bias” in the methods and discussion and ways to mitigate this, however, the concept of “bias” is inconsistent with a qualitative, interpretive approach. Instead, the researchers should apply criteria for rigor that reflect research methods that emphasize the researcher’s role in data collection and analysis and that knowledge is socially constructed.

Yes indeed. This part has been rephrased to avoid the concept of bias. Our axiology paradigm considers researcher values as a perspective on the phenomenon and not a bias: “The impact of the personal values of the main investigator on the data collection process was controlled by a group identification of his preconceived beliefs and by training the investigator in semi-directive interview techniques. Their impact on the interpretive process was limited by a double analysis of every interview and by the diversity of approaches provided by the cross-disciplinary team”. 

We also modified the Methods section to delete mention of a bias: 

“Thematic extraction was triangulated to discuss and enrich interpretation from different perspectives: all interviews were analyzed first by the principal investigator, then by another researcher (GP, psychologist, or sociologist).”

It appears the sample is a convenience rather than purposive sample. Sampling could be strengthened if individuals were recruited based on theoretically informed (rather than merely demographic) criteria. 

We specified that “A convenience sampling method was used to select people with a range of experience in meeting sales reps. No specific inclusion or non-inclusion criteria were imposed since the research team considered that any GP could contribute new data to explain the phenomenon studied. The selection of practitioners was based on variation criteria for the following characteristics: age, gender, rurality, years of practice, patient list size, single or collective practice, and subscription to a paying publication or membership to a peer group (see chart no. 1). […] These characteristics were expected to lead to a diverse sample of experiences and were based on the literature describing how the GP/rep relationship depends on social norms, age, patient list size and the type of continuing medical education.[4,23]”

These characteristics are indeed theoretically informed, based on the Anne Vega report : “These characteristics were expected to lead to a diverse sample of experiences and were based on the literature describing how the GP/rep relationship depends on social norms, age, patient list size and the type of continuing medical education”.

The researchers mention data saturation as a criterion for sampling; however, they do not demonstrate how they knew saturation was achieved and there does not seem to be sufficient presentation of the range of this phenomenon to suggest that they sampled the full range of this experience. This is not a critique of the sample size, but rather, that qualitative researchers must demonstrate to the reader that they sought contrasting perspectives and individuals with diverse experience until no new information arose.

We did specify the process: “Data saturation was discussed and agreed upon by the research team when the sample was considered as gathering a wide range of experiences about relations to sales reps and no new theme emerged from the material analyzed” The variety of profiles is described above in the Methods section (age, sex, type and frequency of contacts with pharma reps, habits about information seeking) 

The researchers stated that this is an ‘exploratory’ analysis, but also seem to try and theorize this phenomenon. Thus, the aims of this work need clarification. If this is more of a theoretical analysis (around hidden curriculum, or cognitive dissonance, e.g. which I think would be of high value), then the theoretical perspectives informing this work and past work on this topic should be brought up front to really highlight what this work adds. At the moment, it is unclear which literature this work is in conversation with and the discussion really contains a lot of useful analysis, but fails to put it into context.

We hope that the modifications mentioned above managed to clarify this point. Our study was exploratory, without any concepts to test: the object was to understand a discrepancy between words and facts. During the research process, participants’ defensive attitudes made the interview guide evolve to a more progressive approach. The cognitive dissonance concept emerged from the inductive analysis. 

At the moment, the analysis remains description, but could be greatly strengthened by incorporating the material in the discussion. The researchers should clarify whether they are applying particular theoretical perspectives, or are conducting a grounded, interpretive analysis (per the stated interpretive phenomenology), and then work to refine the results in either direction.

Citations have been added to illustrate the inductive process. Improvements in the method section are described above. The first paragraph of the discussion has been clarified as follows: In order to understand why some GPs meet sales reps while holding a rather negative opinion of the pharmaceutical industry, an inductive analysis led the authors to categorize the transcript into motivations, representations and values. Furthermore, the data suggested significant sociological and psychological mechanisms that can help explain the reasons for such ambivalence from GPs towards the pharmaceutical industry. The first part of the discussion describes these mechanisms: an experience-based rationale, self-efficacy, cognitive dissonance and a hidden curriculum.

This work is important and I encourage the researchers to continue this work. However, I think it may need to be reworked from the start to really clarify the aim, the theory/methods package, and to re-analyze the findings in this vein.

Thank you for your constructive comments. We hope that the revisions mentioned above achieved clarification.

---

## [Decision Letter · Decision Letter 1]

25 Aug 2021

PONE-D-21-04814R1

General practitioners and sales reps: why are we so ambivalent?

Qualitative study in France comparing perspectives from sociology, psychology and general practice

PLOS ONE

Dear Dr. Barbaroux,

Thank you for submitting your manuscript to PLOS ONE. You have done an excellent job of responding to the comments on the previous review, especially in terms of clarifying the methods that were used. After careful consideration, we feel that it has merit but does not fully meet PLOS ONE’s publication criteria as it currently stands. Therefore, we invite you to submit a revised version of the manuscript that addresses the points raised during the review process.

The manuscript currently reads well but there are still some problems with the English. Joel Lexchin has addressed many of the specific changes needed in his comments. However, the manuscript as a whole would also benefit from being read by a native English-language editor and overall stylistic editing. In some cases, 'false friends' are used - words in English that are similar to French words but have nuanced differences in meaning. An example is the use of 'representations' as a results section header and within the subsequent text. You have explained in the text that this term refers to the physicians' perceptions of the sales representative and of their interactions. However, calling this section 'Physicians' perceptions' would be clearer. When representations is used in the text, this could be better replaced by 'perceptions'.

Secondly, Lisa Parker's recommends a shift to how the content in the results and discussion sections, so that all quotes are included as part of the results, in keeping with standard practices for reporting of qualitative research. She also has some suggestions for organisation of content within sections that would make it easier for the reader to understand the key contrasting themes that surfaced in physicians' interviews.

Thirdly, table 1 would be more readable at a glance if these data were summarised. For example at the top you could just state n=10 and include one column for all GPs, rather than having a separate column per GP, urban setting could be 6 (60%);  Age: Mean (SD); range 30-59; extra fees charged 9 (90%); years in practice mean (SD); range 1-34; enrolled patients mean (SD); range 425-1500; workload (hours per week); mean (SD) etc.

The relevance of some rows (paid and free journal subscriptions) is unclear and for peer group you need an additional description of what the peer group is in the text. I am not sure if these were meant to be indicators of the type of practice the GP has, for example; "Reads commercially-produced free GP journals", or 'Subscribes to La Revue Prescrire' if this is what you were measuring. In the table the row header could be: 'Member of peer GP discussion group'. [or similar] The table should be understandable as a stand-alone item. 

In the comments below, one of Lisa Parker mentions not understanding what you meant by reps not following continuing medical education. I think 'following' may be the problem in this sentence - do you mean that they are not allowed to attend these classes (because of regulations or due to limits imposed by employers?) or that they cannot provide content within continuing education? If this is a comment on a requirement for independence of continuing medical education (e.g. non-sponsorship by industry) this would be an important point.

We look forward to receiving your revised manuscript.

Kind regards,

Barbara Mintzes

Academic Editor

PLOS ONE

Journal Requirements:

Reviewers' comments:

Reviewer's Responses to Questions

**Comments to the Author**

1. If the authors have adequately addressed your comments raised in a previous round of review and you feel that this manuscript is now acceptable for publication, you may indicate that here to bypass the “Comments to the Author” section, enter your conflict of interest statement in the “Confidential to Editor” section, and submit your "Accept" recommendation.

Reviewer #1: (No Response)

Reviewer #4: All comments have been addressed

2. Is the manuscript technically sound, and do the data support the conclusions?

Reviewer #1: Yes

Reviewer #4: Yes

3. Has the statistical analysis been performed appropriately and rigorously? 

Reviewer #1: N/A

Reviewer #4: N/A

4. Have the authors made all data underlying the findings in their manuscript fully available?

Reviewer #1: Yes

Reviewer #4: Yes

5. Is the manuscript presented in an intelligible fashion and written in standard English?

Reviewer #1: No

Reviewer #4: Yes

6. Review Comments to the Author

Reviewer #1: The authors have dealt with my major concerns but there remain a number of minor issues, mostly language related, that need to be corrected.

1. Line 64: Rather than "incite" a better word would be "motivate".

2. The material starting "Cognitive dissonance..." on line 67 to the "hidden curriculum" on line 70 does not belong in the Results. It should be moved to the Discussion.

3. The sentence starting "The strength..." on line 70 should be moved to the end of the Discussion.

4. Line 94: Explain what "direct, indirect, and induced effects" means.

5. Line 97: Delete "prescription" and insert "prescribing".

6. Line 98: Delete "highest" and insert "largest".

7. Line 104; Promotes should start with a lower case p.

8. Line 107: I believe that "follow" should be "fund".

9. Line 129: Delete "prescription" and insert "prescribing".

10. Line 135: When the authors say "second source" do they mean second most frequently used source or second most trusted source?

11. Line 145: Insert "them with" between "provided" and "information".

12. Line 146: Own knowledge of what?

13. Lines 158-159: The authors should explain what they mean by a "constructivist paradigm" and "phenomenological approach".

14. Line 161: When the authors say "this experience" are they referring to interacting with sales reps?

15. Line 190: Use “conference” instead of “congress”.

16. Lines 209-210: What is a "paying publication"?

17. Line 210: Delete "to" and insert "in".

18. Line 213: What do "indemnity" and accreditation mean?

19. Line 220: What is a "thesis survey"?

20. Line 266: Does "verbatims" mean the verbal answers given during the interviews?

21. Line 299: What sales rep budget restrictions are the authors referring to?

22. Line 364: What do the authors mean by "reassessment"?

23. Line 371: Substitute "beliefs" for "cognition".

24. Line 407: Use "companies" instead of "industries".

25. Line 422: Substitute "observations" for "transcriptions".

26. Line 438: Sales visit charter has previously been capitalized.

27. Line 487: It should be “et al”.

28. Line 499: Use "conferences" instead of "congresses".

Reviewer #4: GPs and sales reps

Synopsis – this paper reports on an empirical qualitative study, involving interviews with GPs in France. The study focus is to explore the phenomenon whereby GPs regularly meet with sales reps, despite holding negative views about the pharmaceutical industry and sales rep marketing.

Overall comment - This is an important topic and empirical work is always a welcome addition to the literature. The current study contains useful data but could be greatly strengthened by further / clearer presentation of the analysis.

Major comments

I would have liked to see clearer presentation of the analytic work. At present it seems that the Results section is very descriptive, and the stronger analysis is buried in the Discussion section. I suggest re-ordering some of the text to incorporate the analytic work into Results. This would also adhere more closely to the standard results/discussion format of academic papers in the biomedical sector, where quotes etc are generally not placed in Discussion. For example, the authors could consider sectioning the Results into [a] negative views about sales reps [b] reasons for meeting with sales reps (need for quick information; valuing experience-based knowledge; habits acquired from medical school days via the hidden curriculum) [c] how GPs manage the cognitive dissonance (e.g. feeling that self is invulnerable to sales rep tactics etc.) Of course the authors do not have to follow this particular analytic structure, this is just one possible way that the analysis could be taken further to address the stated aim of trying to understand why this phenomenon is happening and what it’s like for the GPs on the ground. I see that the authors have already done a substantial revision in response to previous reviewer comments. Nevertheless, I think a further revision would greatly strengthen the paper.

Minor comments

Intro

p 5 line 103 – ‘the information provided by sales reps remains poor’ – what information and who is it provided to ?

P 5 line 107 ‘pharma sales reps are not allowed to follow continuing medical education courses.’ This comment is not clear to me, what do you mean?

P6 line 137 – ‘this situation is problematic on a financial level … and on a social, ethical and public health level.’ I probably agree with you, but I’d like some more detail here, can you spell out why it is problematic in each of these contexts?

Methods

P 8 line 181 was the preliminary interview data included in this study?

P 10 line 222 you talk about saturation here, but earlier you said that you were not aiming for saturation and it was not needed in a phenomenological study

P 10 line 230 I would like more information about why ethics approval was not required. Certainly in my country any studies involving human participants requires ethics approval, even when it is a low-risk study such as this one.

Results

P 11 line 253 it’s not useful to give 118 pages of transcriptions unless you provide the font, margins, line spacing etc; you could provide word count? Probably unnecessary; the number and average / median duration of interviews is sufficient I think

Strengths and limitations

P 19 line 446 small number of interviews is not always a limitation of a qualitative study, typically this is how qualitative research works. It would only be a limitation if you were unable to complete as many studies as you wanted to, e.g. if funding ran out or other reasons. In addition it is not always expected to get participants to review their transcripts, and I wouldn’t think this necessarily constituted a limitation unless there were particular reasons e.g. if the participants had strong accents and you were worried about transcribing their words correctly.

7. PLOS authors have the option to publish the peer review history of their article (what does this mean?). If published, this will include your full peer review and any attached files.

Reviewer #1: **Yes: **Joel Lexchin

Reviewer #4: **Yes: **Lisa Parker

---

## [Author Response · Author response to Decision Letter 1]

8 Oct 2021

Response to reviewers:

Dear Barbara Mintzes, Academic Editor,

Dear Joel Lexchin and Lisa Parker , reviewers,

Thank you for giving us the opportunity to improve our manuscript once again. We have considered all your comments carefully. Please find below our point-by-point response.

Academic Editor reports:

The manuscript currently reads well but there are still some problems with the English. Joel Lexchin has addressed many of the specific changes needed in his comments. However, the manuscript as a whole would also benefit from being read by a native English-language editor and overall stylistic editing. In some cases, 'false friends' are used - words in English that are similar to French words but have nuanced differences in meaning. An example is the use of 'representations' as a results section header and within the subsequent text. You have explained in the text that this term refers to the physicians' perceptions of the sales representative and of their interactions. However, calling this section 'Physicians' perceptions' would be clearer. When representations is used in the text, this could be better replaced by 'perceptions'.

Thank you for this suggestion. A second native reviewer did a final polish of the English. All the changes suggested by Joel Lexchin have been taken into consideration and each occurrence of “representations” have been replaced with “physicians’ perceptions”.

Secondly, Lisa Parker's recommends a shift to how the content in the results and discussion sections, so that all quotes are included as part of the results, in keeping with standard practices for reporting of qualitative research. She also has some suggestions for organisation of content within sections that would make it easier for the reader to understand the key contrasting themes that surfaced in physicians' interviews.

Done (details below in the answer to the reviewer)

Thirdly, table 1 would be more readable at a glance if these data were summarised. For example at the top you could just state n=10 and include one column for all GPs, rather than having a separate column per GP, urban setting could be 6 (60%); Age: Mean (SD); range 30-59; extra fees charged 9 (90%); years in practice mean (SD); range 1-34; enrolled patients mean (SD); range 425-1500; workload (hours per week); mean (SD) etc.

Done. The table now looks like this : 

Table 1. Characteristics of the ten GPs interviewed.

 N = 10

 (Mean ± SD [Range])

Age (years) 48,8 ± 9 [30-59]

Years in practice 15 ± 11 [1-34]

Enrolled patients 906 ± 439 [400-1500]

Hours p. week workload 55 ± 8 [43-65]

Number of sales reps p. week 3 ± 3 [0-10]

 (n - percentage)

Extra fee charged 1 (10%)

Urban setting 6 (60%)

Group practice 5 (50%)

Subscribes to a paid medical journal 1 (10%)

Reads commercially-produced free GP journals 4 (40%)

Member of GP peer group 3 (30%)

The relevance of some rows (paid and free journal subscriptions) is unclear and for peer group you need an additional description of what the peer group is in the text. I am not sure if these were meant to be indicators of the type of practice the GP has, for example; "Reads commercially-produced free GP journals", or 'Subscribes to La Revue Prescrire' if this is what you were measuring. In the table the row header could be: 'Member of peer GP discussion group'. [or similar] The table should be understandable as a stand-alone item.

Done.

In the comments below, one of Lisa Parker mentions not understanding what you meant by reps not following continuing medical education. I think 'following' may be the problem in this sentence - do you mean that they are not allowed to attend these classes (because of regulations or due to limits imposed by employers?) or that they cannot provide content within continuing education? If this is a comment on a requirement for independence of continuing medical education (e.g. non-sponsorship by industry) this would be an important point.

Indeed, Pharma sales reps are not allowed to provide content for continuing education courses. This sentence was clarified in the text.

Reviewers' comments:

Reviewer #1: Joel Lexchin

The authors have dealt with my major concerns but there remain a number of minor issues, mostly language related, that need to be corrected.

1. Line 64: Rather than "incite" a better word would be "motivate".

Done

2. The material starting "Cognitive dissonance..." on line 67 to the "hidden curriculum" on line 70 does not belong in the Results. It should be moved to the Discussion.

Done

3. The sentence starting "The strength..." on line 70 should be moved to the end of the Discussion.

Done

4. Line 94: Explain what "direct, indirect, and induced effects" means.

Done : “In the United States of America (USA), the biopharmaceutical sector generates nearly $1.2 trillion in economic output annually when direct (biopharmaceutical industry revenue), indirect (vendor and supplier activity), and induced effects (additional private economic activity) are considered(1)”.

5. Line 97: Delete "prescription" and insert "prescribing".

Done

6. Line 98: Delete "highest" and insert "largest".

Done

7. Line 104; Promotes should start with a lower case p.

Done

8. Line 107: I believe that "follow" should be "fund".

Modified as follow : “Pharma sales reps are not allowed to provide content within continuing education courses”.

9. Line 129: Delete "prescription" and insert "prescribing".

Done

10. Line 135: When the authors say "second source" do they mean second most frequently used source or second most trusted source?

The report concluded that, after the drug database Vidal, sales rep visits were the second most used source of information (modified)

11. Line 145: Insert "them with" between "provided" and "information".

Done

12. Line 146: Own knowledge of what?

GPs also described a lack of confidence in their own pharmaceutical knowledge when leaving university (modified)

13. Lines 158-159: The authors should explain what they mean by a "constructivist paradigm" and "phenomenological approach".

Done : “An exploratory qualitative study was conducted based on a constructivist paradigm.[25] In this paradigm, qualitative research methods are considered as a means to allow the investigator to be the primary interpretive instrument and reality is considered as socially constructed and knowledge as a product constructed by people taking active part in a research process.[25] An interpretative phenomenological approach was used, that is, individual professional practices were explored and the phenomenon was modeled through the in-depth analysis of singular experiences, the meaning that participants gave to their experience of meeting sales reps, and the underlying psychological mechanisms.[26] The purpose of a phenomenological approach is mainly to shed light on the essence of a person’s experience in relation to a specific phenomenon, in this instance meeting sales reps despite holding a negative opinion of them.”

14. Line 161: When the authors say "this experience" are they referring to interacting with sales reps?

Yes (modified)

15. Line 190: Use “conference” instead of “congress”.

Done

16. Lines 209-210: What is a "paying publication"?

We meant a journal which charges a fee for subscription, this has been modified.

17. Line 210: Delete "to" and insert "in".

Done

18. Line 213: What do "indemnity" and accreditation mean?

Modified as follows : “can give rise to a government payment to members when the group is accredited (registered and recognized by the government)”

19. Line 220: What is a "thesis survey"?

We modified this sentence as follows : “The GPs were asked to answer a survey as part of an end of course medical thesis. The survey was presented to them as dealing with the information sources used by general practitioners, without mentioning promotional detailing visits.”

20. Line 266: Does "verbatims" mean the verbal answers given during the interviews?

In qualitative research, a transcript or verbatim is defined as a textual unit extracted from the transcription of the verbal answers given during the interviews, that can be smaller than a sentence. This has been clarified in the manuscript : “Transcripts, or excerpts of responses, can be grouped into the following categories: motivations, physicians’ perceptions, and values”.

21. Line 299: What sales rep budget restrictions are the authors referring to?

The sentence was unclear: participants referred to budget restrictions that have reduced the influence of the pharmaceutical industry. The sentence was reworded as follows: “The pharmaceutical industry’s persuasive force was described as belonging to a past era because current sales rep budget restrictions mean fewer reps and gifts: “There’s no more! There’s nothing left!””

22. Line 364: What do the authors mean by "reassessment"?

We meant self-questioning. Modified.

23. Line 371: Substitute "beliefs" for "cognition".

Done

24. Line 407: Use "companies" instead of "industries".

Done

25. Line 422: Substitute "observations" for "transcriptions".

Done

26. Line 438: Sales visit charter has previously been capitalized.

Thank you for noting this typo.

27. Line 487: It should be “et al”.

Thank you for noting this typo.

28. Line 499: Use "conferences" instead of "congresses".

Done.

Reviewer #4: Lisa Parker

Major comments

I would have liked to see clearer presentation of the analytic work. At present it seems that the Results section is very descriptive, and the stronger analysis is buried in the Discussion section. I suggest re-ordering some of the text to incorporate the analytic work into Results. This would also adhere more closely to the standard results/discussion format of academic papers in the biomedical sector, where quotes etc are generally not placed in Discussion. 

Thank you for this comment, a former reviewer commented that “At the moment, the analysis remains description, but could 

be greatly strengthened by incorporating the material in the discussion.”, which is the reason we added quotes in the discussion. Following your advice, quotes have been removed from the discussion, as in our former submission.

For example, the authors could consider sectioning the Results into [a] negative views about sales reps [b] reasons for meeting with sales reps (need for quick information; valuing experience-based knowledge; habits acquired from medical school days via the hidden curriculum) [c] how GPs manage the cognitive dissonance (e.g. feeling that self is invulnerable to sales rep tactics etc.) Of course the authors do not have to follow this particular analytic structure, this is just one possible way that the analysis could be taken further to address the stated aim of trying to understand why this phenomenon is happening and what it’s like for the GPs on the ground. I see that the authors have already done a substantial revision in response to previous reviewer comments. Nevertheless, I think a further revision would greatly strengthen the paper.

Thank you for this comment. We modified the structure of the paper to emphasize the concepts and psychosocial mechanisms involved in dealing with ambivalence: they were integrated in the results section, with dedicated subsections for clarity, and we reduced the repetition concerning pragmatism and evidence-based rationale. The hidden curriculum is still in the discussion section, as it relied less on data and can be considered more as explicative of the experience and perspective of participants. 

There are no longer quotes in the discussion section.

Minor comments

Intro

p 5 line 103 – ‘the information provided by sales reps remains poor’ – what information and who is it provided to ?

Thank you for this comment, we modified this sentence as follows: “the information concerning drugs provided by sales reps to GPs remains poor”

P 5 line 107 ‘pharma sales reps are not allowed to follow continuing medical education courses.’ This comment is not clear to me, what do you mean?

Pharma sales reps are not allowed to provide content within continuing education courses. (modified)

P6 line 137 – ‘this situation is problematic on a financial level … and on a social, ethical and public health level.’ I probably agree with you, but I’d like some more detail here, can you spell out why it is problematic in each of these contexts?

We clarified as follows: “as much as on a social and ethical level (citizens’ demand for more transparency and more independent training for prescribers) and on a public health level because of potentially inadequate prescribing practices.(ref)”

Methods

P 8 line 181 was the preliminary interview data included in this study?

Yes. The data from the preliminary interviews was retained in this study, in accordance with the standards in qualitative research (27,29). (added)

P 10 line 222 you talk about saturation here, but earlier you said that you were not aiming for saturation and it was not needed in a phenomenological study

Indeed, the wording of the sentence could be considered confusing and has been removed. We left the following sentence in the text as a strength concerning the analysis process : “The research team discussed the data sample and considered that it gathered a wide range of experiences about relationships with sales reps and no new theme emerged from the material analyzed”.

P 10 line 230 I would like more information about why ethics approval was not required. Certainly in my country any studies involving human participants requires ethics approval, even when it is a low-risk study such as this one.

Data was collected before January 2016 in France. At this time, ethics approval was needed only for pharmaceutical research. In January 2016, the French “loi jardé” made ethics approval mandatory for some other kinds of research, such as non-pharmaceutical research involving patients. Currently, there is still no legal obligation to ask for ethics approval for this kind of research since the law does not consider qualitative research as research involving human participants. You can find out more at : 

https://www.legifrance.gouv.fr/jorf/id/JORFTEXT000034634217

« Ne sont pas des recherches impliquant la personne humaine au sens du présent titre les recherches et qui visent à évaluer des modalités d'exercice des professionnels de santé ou des pratiques d'enseignement dans le domaine de la santé. » that we can translate as « Research involving the human person within the meaning of this title does not include research aimed at evaluating the methods of practice of health professionals or teaching practices in the health field. »

Results

P 11 line 253 it’s not useful to give 118 pages of transcriptions unless you provide the font, margins, line spacing etc; you could provide word count? Probably unnecessary; the number and average / median duration of interviews is sufficient I think

We agree with you, this information has been deleted.

Strengths and limitations

P 19 line 446 small number of interviews is not always a limitation of a qualitative study, typically this is how qualitative research works. It would only be a limitation if you were unable to complete as many studies as you wanted to, e.g. if funding ran out or other reasons. In addition it is not always expected to get participants to review their transcripts, and I wouldn’t think this necessarily constituted a limitation unless there were particular reasons e.g. if the participants had strong accents and you were worried about transcribing their words correctly.

Thank you for this comment, we do agree with you. This paragraph was reworded in order not to present it as a limitation.

The research team would like to thank the editor and the reviewers again for the quality of their reviews.

We look forward to hearing from you regarding our revised submission.

Sincerely,

Dr Adriaan BARBAROUX

On behalf of the research team.

---

## [Decision Letter · Decision Letter 2]

9 Dec 2021

General practitioners and sales reps: why are we so ambivalent?

Qualitative study in France comparing perspectives from sociology, psychology and general practice

PONE-D-21-04814R2

Dear Dr. Barbaroux,

We’re pleased to inform you that your manuscript has been judged scientifically suitable for publication and will be formally accepted for publication once it meets all outstanding technical requirements.

Kind regards,

Barbara Mintzes

Academic Editor

PLOS ONE

Additional Editor Comments (optional):

I have an issue I would like to raise with you concerning data availability:

**Supplementary file 1** was very interesting and includes some fantastic quotes beyond those in the article and the flavour of the discussion comes out strongly in the original French. Your openness in making it available publicly is also laudable. However, I am concerned that there are too many potentially identifying details within these transcripts, including interviewees mentioning names or places. It am concerned about breach of privacy.

**My recommendations would be:**

1) to ensure that making these transcripts available is consistent with the degree of protection of privacy offered to participants on the consent form.

2) If you would like to make the file available, I would also suggest that you only make this file available on request to other researchers and that before providing it, you read through all of the interviews and **remove all potentially identifying information.**

At my university, we are subject to ethics approval of interview studies and all transcripts must be kept confidential, sensitive or not. I also understand that there is strong legislated protection of personal privacy in the EU. Please ensure that any final data sharing decision is consistent with legal obligations. 

Additionally, usually if a data file such as this was made available, it would only be available as ‘raw data’ with no highlighting or other kinds of first-pass analysis.

I also wanted to discuss the reviewers' comments. Joel Lexchin has made two main suggestions:

Shifting in the placement of large blocks text from results to discussion in several places;Copy-editing that is still needed.

My suggestion is that you look at his recommended changes in placement of text. If you feel that these shifts would strengthen the paper, please go ahead, but if not, it is fine for you to maintain the current organisation.

Some of the text in the results section would normally be included in the discussion in a medical journal article, but your explanatory text helps to frame the quotes in the results section and supports the approach to re-organisation that Lisa Parker suggested at a last stage, and that you have done an excellent job of implementing. I find the article to currently be very informative and readable and am happy to maintain the current structure, especially given that there is some extra flexibility in a report of a qualitative interview study.

For copy-editing, please do incorporate Joel lexchin’s recommendations:

1. It should be USA not US.

2. Line 174: “on” should be inserted after “commented”.

3. Line 213: “rurality” is not a word, use something like “rural location”.

4. Line 273: does "it" refer to the information received from sales reps or meeting sales reps?

5. Line 301: it should be "on" rather than "to".

I also had a few copy-editing suggestions:

The title should be:

"General practitioners and pharmaceutical sales representatives: why are we so ambivalent?"*[remove the abbreviation ‘sales reps’]*

Page 5, lines 104-105:

Despite strict regulations (health authorities can impose fines of up to €10,000 or 10% of a product’s 105 annual sales revenue) [7], the Sales Visit Charter is rarely respected.[5,6]

instead: “key provisions within the Sales Visit Charter, such as….  are rarely respected.” [*e.g. please avoid overstating the case as administrative and certification policies are respected.], *

P6, line 139: British GPs received visits. This should be, “British GPs reported that they received visits from sales reps for…”

P9, line 196: “…was chosen because it is in a situation of doubt…”

This should be: “…was chosen because of previous evidence that in a situation of doubt…”

P10, line 233:

“…and the quotations are not identified.” Perhaps instead: “…and no personal characteristics such as gender that might identify a speaker are attached to specific quotes.”

P11, Lines 236-7: “The authors declare that they did not receive …” should be, “The authors did not receive any funding for this study.”

P12, Table 1: Please left justify the text on the table left column and use rows that span the entire width of the table, including header rows. Please add a heading such as ‘demographics’ and ‘practice characteristics’ above the relevant sections.  I would suggest putting the table at the end of the manuscript, with a note such as [table 1 here] if you would like to specify its location in the txt. A table can be single spaced for ease of reading and the font size can be sans serif but should not be larger than the usual text font.

‘Extra fee charged’ should be “Patients charged extra fees”.  I would also suggest generally listing characteristics from the most frequent to least frequent (except that types of journals should be together). ‘p. week’ should be ‘per week’.

P12, line 265: “physician did not address the visit spontaneously, the interviewer raised the subject specifically” – please change ‘visit’ to ‘issue’.

P13, line 273, ”receiving sales rep visits despite their negative opinion about it.” – please change “negative opinion about it.” To “negative opinion about these interactions.”

Organizational problems that need to be corrected: 1. The material from lines 334-339 should be moved to the Discussion. 2. The material from lines 344-349 should be moved to the Discussion. 3. The material from lines 356-374 does not belong in the Results, except for the quotation on lines 364-366.

Reviewers' comments:

Reviewer's Responses to Questions

**Comments to the Author**

1. If the authors have adequately addressed your comments raised in a previous round of review and you feel that this manuscript is now acceptable for publication, you may indicate that here to bypass the “Comments to the Author” section, enter your conflict of interest statement in the “Confidential to Editor” section, and submit your "Accept" recommendation.

Reviewer #1: (No Response)

Reviewer #4: All comments have been addressed

2. Is the manuscript technically sound, and do the data support the conclusions?

Reviewer #1: Yes

Reviewer #4: Yes

3. Has the statistical analysis been performed appropriately and rigorously? 

Reviewer #1: N/A

Reviewer #4: N/A

4. Have the authors made all data underlying the findings in their manuscript fully available?

Reviewer #1: Yes

Reviewer #4: No

5. Is the manuscript presented in an intelligible fashion and written in standard English?

Reviewer #1: Yes

Reviewer #4: Yes

6. Review Comments to the Author

Reviewer #1: There are some organizational problems that need to be corrected:

1. The material from lines 334-339 should be moved to the Discussion.

2. The material from lines 344-349 should be moved to the Discussion.

3. The material from lines 356-374 does not belong in the Results, except for the quotation on lines 364-366.

There are also a few continuing copy editing issues:

1. It should be USA not US.

2. Line 174: “on” should be inserted after “commented”.

3. Line 213: “rurality” is not a word, use something like “rural location”.

4. Line 273: does "it" refer to the information received from sales reps or meeting sales reps?

5. Line 301: it should be "on" rather than "to".

Reviewer #4: (No Response)

7. PLOS authors have the option to publish the peer review history of their article (what does this mean?). If published, this will include your full peer review and any attached files.

Reviewer #1: **Yes: **Joel Lexchin

Reviewer #4: **Yes: **Lisa Parker

---

## [Editor Report · Acceptance letter]

12 Jan 2022

PONE-D-21-04814R2 

**General practitioners and sales representatives: why are we so ambivalent?**
Qualitative study in France comparing perspectives from sociology, psychology and general practice 

Dear Dr. Barbaroux:

I'm pleased to inform you that your manuscript has been deemed suitable for publication in PLOS ONE. Congratulations! Your manuscript is now with our production department. 

Kind regards, 

on behalf of

Dr. Barbara Mintzes 

Academic Editor

PLOS ONE